

# Regional uncertainty of GOSAT XCO$_2$ retrievals in China: Quantification and attribution

Nian Bie[1,2], Liping Lei[1], Zhaocheng Zeng[3], Bofeng Cai[4], Shaoyuan Yang[1,2], Zhonghua He[1,2], Changjiang Wu[1,2], and Ray Nassar[5]

[1]Key Laboratory of Digital Earth Science, Institute of Remote Sensing and Digital Earth, Chinese Academy of Sciences, Beijing 100094, China
[2]University of Chinese Academy of Sciences, Beijing 100049, China
[3]Division of Geological and Planetary Sciences, California Institute of Technology, Pasadena, CA91125, USA
[4]The Center for Climate Change and Environmental Policy, Chinese Academy for Environmental Planning, Ministry of Environmental Protection, Beijing, 100012, China
[5]Climate Research Division, Environment and Climate Change Canada

*Correspondence to*: leilp@radi.ac.cn

**Abstract**. The regional uncertainty of XCO$_2$ (column-averaged dry air mole fraction of CO$_2$) retrieved using different algorithms from the Greenhouse gases Observing SATellite (GOSAT) and its attribution are still not well understood. This paper investigates the regional performance of XCO$_2$ within a band of 37°N~ 42°N segmented into 8 cells in a grid of 5° from west to east (80°E ~120°E) in China, where there are typical land surface types and geographic conditions. The former include the various land covers of desert, grassland and built-up areas mixed with cropland, and the latter include anthropogenic emissions that tend to be small to large from west to east, including those from the megacity of Beijing. For these specific cells, we evaluate the regional uncertainty of GOSAT XCO$_2$ retrievals by quantifying and attributing the consistency of XCO$_2$ retrievals from five algorithms (ACOS, NIES, EMMA, OCFP, and SRFP) by intercomparison and particularly by comparing these with simulated XCO$_2$ from the Goddard Earth Observing System 3-D chemical transport model (GEOS-Chem), the nested model in East Asia. We introduce the anthropogenic CO$_2$ emissions data generated from the investigation of surface emitting point sources that was conducted by the Ministry of Environmental Protection of China to GEOS-Chem simulations of XCO$_2$ over the Chinese mainland. The results indicate that (1) regionally, the five algorithms demonstrate smaller absolute biases between 0.9-1.5 ppm in eastern cells, which are covered by built-up areas mixed with cropland with intensive anthropogenic emissions, than those in the western desert cells with a high-brightness surface, 1.2-2.2 ppm from the pairwise comparison results of XCO$_2$ retrievals. The inconsistency of XCO$_2$ from the five algorithms tends to be high in the Taklimakan Desert in western cells, which is likely induced by high surface albedo in addition to dust aerosols in this region. (2) Compared with XCO$_2$ simulated by GEOS-Chem (GEOS-XCO$_2$), the XCO$_2$ values of ACOS and SRFP better agree with GEOS-XCO$_2$, while OCFP is the least consistent with GEOS-XCO$_2$. (3) Viewing attributions of XCO$_2$ in the spatio-temporal pattern, ACOS, SRFP and EMMA demonstrate similar patterns, while OCFP is largely different from the others. In conclusion, the discrepancy in the five algorithms is the smallest in eastern cells in the investigated band where the megacity of Beijing is located and where there are strong anthropogenic CO$_2$ emissions, which




implies that $XCO_2$ from satellite observations could be reliably applied in the assessment of atmospheric $CO_2$ enhancements
induced by anthropogenic $CO_2$ emissions. The large inconsistency among the five algorithms presented in western deserts
with a high albedo and dust aerosols, moreover, demonstrates that further improvement is still necessary in such regions,
even though many algorithms have endeavored to minimize the effects of aerosols and albedo.

Key words: GOSAT, $XCO_2$ retrieval algorithms, simulated $XCO_2$ by GEOS-Chem, regional uncertainty, anthropogenic
emission, and desert
**1 Introduction**
The column-averaged dry air mole fraction of $CO_2$ ($XCO_2$) derived from satellite observations, such as the SCanning
Imaging Absorption spectroMeter of Atmospheric CHartographY (SCIAMACHY ) (Burrows et al., 1995; Bovensmann et al.,
1999) , the Greenhouse gases Observing SATellite (GOSAT) (Yokoda et al., 2004), Orbiting Carbon Observatory (OCO-2)
(Crisp et al., 2004), and Chinese Carbon Satellite (TanSat) (Liu et al., 2013), have largely improved our understanding of
the variation in atmospheric $CO_2$ concentration and carbon sources and sinks at a global and regional scale. There have been
several full-physics retrieval algorithms specially developed for retrieving $XCO_2$ from the GOSAT spectrum, mainly
including the NASA Atmospheric $CO_2$ Observations from Space (ACOS) (O'Dell et al., 2012), the National Institute for
Environmental Studies (NIES) (Yoshida et al., 2013), University of Leicester full-physics $XCO_2$ (OCFP) (Cogan et al., 2012)
and RemoTeC $XCO_2$ Full Physics (SRFP) (Butz et al., 2011). Additionally, the ensemble median algorithm EMMA was put
forward as a combination of retrieval products from independent algorithms including ACOS, NIES, OCFP, and SRFP
(Reuter et al., 2013).
Satellite-retrieved $XCO_2$ is susceptible to the effects of the light path, observed spectrum, surface states, and so
on(O'Dell et al., 2012; Oshchepkov et al., 2013).The bias and performance of $XCO_2$ from an algorithm could change in
different regions with differing land surfaces and anthropogenic emissions. Spatio-pattern attributions of $XCO_2$ viewed from
different algorithms are also different, even in the same region, due to the different physical approaches of the algorithms,
assumptions of atmospheric conditions (aerosol, surface pressure, $CO_2$ profile, etc.), and pre- and post-processing filters. To
date, the validation of $XCO_2$ retrievals from these algorithms focuses on using ground-based measurements from Total
Carbon Column Observing Network (TCCON) sites (Wunch et al., 2011; Yoshida et al., 2013; Hewson, 2016; Buchwitz et
al., 2015, Detmers et al., 2015, Oshchepkov et al., 2013) and their consistency evaluation and cross-comparison both at a
global scale and in continental regions (Kulawik et al., 2016; Lindqvist et al., 2015; Lei et al., 2014).The precision and
uncertainty of satellite-retrieved $XCO_2$ outside TCCON stations, most of which are located remote from abundant biosphere
fluxes and human activities, are still not well evaluated. The sparseness of TCCON stations over the globe, moreover, means
a lack of enough ground observations to validate satellite retrievals. Specifically, there are no good TCCON data available in
China, and only a few satellite retrievals were validated using ground-based Fourier Transform Spectrometer (FTS) $XCO_2$





measurements in Hefei (Wang et al., 2016). In the analysis and application of XCO$_2$ data from ACOS, NIES, OCFP, SRFP
and EMMA, we found that unreasonably high XCO$_2$ was demonstrated in the Taklimakan desert in China (Bie et al., 2016;
Liu et al., 2015). For this reason, we extended the scope to select a larger study period and to further assess the overall
performance of these five algorithms at a regional scale.
With the advantage of continuity in space and time, atmospheric transport model simulation of CO$_2$ has been widely
used in assessing the performance of satellite-retrieved XCO$_2$ (Cogan et al., 2012; Lindqvist et al., 2015; Kulawik et al.,
2016). As anthropogenic emission of CO$_2$ is the major contributor to increases of CO$_2$ in the atmosphere, many studies have
been involved in deriving estimates of anthropogenic CO$_2$ emissions (Oda et al., 2011; Andres et al., 2011). It is known that
there exists high uncertainty in estimates of CO$_2$ emissions from both the burning of fossil fuel and cement production (FF
CO$_2$ emissions) throughout China (Guan et al., 2012; Liu et al., 2015). As noted by Andrews et al. (2012), there exist many
kinds of restrictions (e.g., commercial competitiveness reasons) in obtaining accurate data on sub-national (e.g., large-point-
source or provincial) FF CO$_2$ emissions. Furthermore, the assumption of uniform per-capita emissions within a country has
also been shown to be unreliable for large countries with diversified economies and electricity-generation methods (Nassar et
al., 2013). In the previous study of Keppel-Aleks (2013), the simulated Chinese XCO$_2$ data was increased by a national
uniform ratio for the corresponding XCO$_2$ contributed by fossil sources to account for the underestimation in Chinese
emissions, in which way the spatial variability of Chinese FF emissions was not considered sufficient.
In this paper, we focus on a latitude band of 37°N-42°N from 80°E to 120°E in China, where there are various typical
land covers such as desert, including the Taklimakan desert, and grassland and built-up areas mixed with croplands,
including the megacity of Beijing, and there are anthropogenic emissions that trend from small amounts to large amounts
from west to east. In this band, the inconsistencies of XCO$_2$ values derived from five algorithms including ACOS V3.5,
NIES V02.21, OCFP 6.0, SRFP V2.3.7, and EMMA V2.1c are compared and evaluated in this paper. Moreover, a forward
model simulation data set from GEOS-Chem is also used for intercomparison. To improve the simulation of CO2
concentration, we introduced a new emission data set, the Chinese High Resolution Emission Gridded Data (CHRED),
which is produced by the Ministry of Environmental Protection, China (MEP) based on investigations of emitting point
sources from approximately 150 million enterprises throughout the country in 2012 (Wang et al., 2014; Cai et al., 2014).
First, we aim to reveal the regional uncertainty of XCO$_2$ observed by GOSAT for the different land covers and
anthropogenic CO$_2$ emission regions through the inconsistency of five algorithms, and second, we aim to give a reasonable
and valuable reference for the analysis and application of XCO$_2$ data when using these XCO$_2$ data from the five algorithms.
Sec. 2 in this paper describes the XCO$_2$ retrievals data from five algorithms and the implementation of XCO$_2$ simulated by
GEOS-Chem using CHRED. Inconsistencies of XCO$_2$ datasets among the five algorithms are quantified and evaluated as
follows: pairwise comparisons of XCO$_2$ between algorithms and comparisons with GEOS-Chem simulations in Sec. 3. The
spatio-temporal patterns of XCO$_2$ from each algorithm are investigated using a combination of sine and cosine trigonometric
functions to fit monthly averaged XCO$_2$ from March 2010 to February 2013 in Sec. 4. Furthermore, the most likely
attribution-affecting factors on regional inconsistency, including aerosol and surface albedo, are described in Sec. 5. The



latest ACOS V7.3 dataset, moreover, is also evaluated by cross-comparisons with GEOS-Chem and other algorithms
including ACOS V3.5, NIESV02.XX, OCFP V6.0, SRFP V2.3.7 and EMMA v2.1.C, as shown in subsections of Sec. 5.
Finally, the regional performances of five algorithms and the regional uncertainty of GOSAT XCO2 retrievals from the
results above are summarized, and conclusions are given in Sec. 6.

## 2 Study area and data

### 2.1 Study area

The latitude band of 37°N~42°N from 80°E to 120°E in China is selected as the study area, which is segmented into eight
cells in a grid of 5 °×5 ° units for comparison and evaluation. The study area has two typical surface characteristics as shown
in Fig. 1, supporting our assessment of the performance of $XCO_2$ retrievals from five algorithms: (1) the difference of
anthropogenic $CO_2$ emissions from west to east is significant going from small amounts to large amounts as shown in Fig.
1(left), where data are from the Open-source Data Inventory for Anthropogenic $CO_2$ (ODIAC), a global annual fossil fuel
$CO_2$ emission inventory developed by combining a worldwide point-source database and satellite observations of the global
nightlight distribution (Oda et al., 2011). There are almost no anthropogenic $CO_2$ emissions in the western cells ending at
105 °E, while there is high anthropogenic emission located in the cells on the eastern end. (2) There are typical land covers
from west to east mainly composed of desert (sand in two cells from 80 °E to 90 °E, Gobi in two cells from 90 °E to 100 °E,
sand in a cell from 100 °E to 105 °E), grassland in a cell from 105 to 110 °E, and cropland and built-up areas in two cells from
110 °E to 120 °E as shown in Fig. 1 (right). These characteristics may bring about complicated aerosol composition and
concentration. One of the main reasons for focusing on this band, however, is the greater availability of high-quality GOSAT
scans in this area compared to other areas in China.

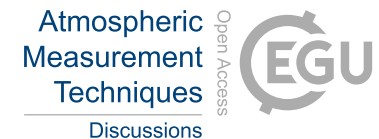

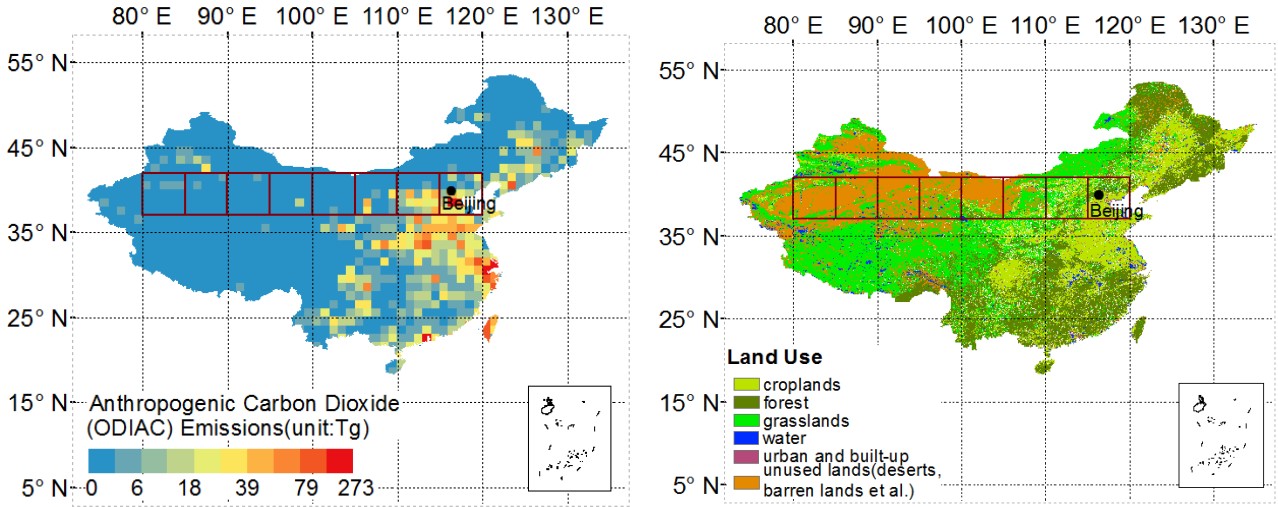

**Fig. 1. Location of the study area segmented into cells (deep red cells) in China and annual fossil fuel CO$_2$ emission in 2012 (left) (1**
**x 1 degree) from ODIAC and land use mapping (right) in 2010, where the black dot represents Beijing, the capital of China.**

**2.2 GOSAT XCO$_2$ dataset derived from five algorithms**
We collected XCO$_2$ data from March 2010 to February 2013 derived from five algorithms: ACOS V3.5
(http://CO$_2$.jpl.nasa.gov), NIES V02.xx (RA version with GU screening scheme) (http://data.gosat.nies.go.jp/
GosatUserInterfaceGateway/ guig/GuigPage/ open.do?l ang=en), OCFP V6.0, SRFP V2.3.7 and EMMA V2.1c (http://www.
esa- ghg-cci.org/ sites/ default/ files/ /documents/public/documents/GHG-CCI_DATA.html) , the version of EMMA without
SCIAMACHY data included. The major characteristics of the five algorithms and the data sources are listed in Table 1. The
validation at TCCON sites for all algorithms indicates that the bias is less than 1.2 ppm and that the standard deviation is less
than 2.0 ppm. All algorithms take aerosol optical depth (AOD) into consideration in their data screening scheme but in
slightly different manners. The recommended bias corrections are applied to the collected XCO$_2$ data from ACOS, OCFP
and SRFP . Data observed with high gain and passing the corresponding recommended quality control are used in ACOS,
NIES, OCFP and SRFP while all data from EMMA are used.



**Table 1 Summary of validating results, data screening schemes, bias corrections and consideration in scattering among algorithms.**

| | ACOS | NIES | OCFP | SRFP | EMMA |
|---|---|---|---|---|---|
| Validation with TCCON* | 0.3 ppm<br>1.7 ppm | -1.2 ppm<br>2.0 ppm | 0.04 ppm<br>1.78ppm | 0.01 ppm<br>1.93 ppm | 0.28 ppm<br>1.9 ppm |
| Data screening scheme | Aerosol_total_aod: 0.015 to 0.25<br>Sounding_altitude:<3000<br>$0.55<XCO_2$_uncer$<2.0$ ppm<br>Aod_dust<0.15<br>The difference of the retrieved and priori surface pressure from the A-band cloud-screen $\Delta P_{s,cld}$ :(-12,4.1) hPa | Retrieved aerosol optical thickness:<=0.1<br>Difference of retrieved and a priori surface pressure:<=20 hpa<br>Blended albedo: <1 | Retrieved type 1 (small) AOD: <=0.3<br>Retrieved type 2 (large) AOD: <=0.15<br>Retrieved ice type AOD: <=0.025<br>Error on retrieved $XCO_2$ :<=2.15 | Aerosol optical thickness :<0.3<br>3<aero_size<5<br>0<aerosol_filter<300<br>Error on retrieved $XCO_2$: <1.2 ppm<br>standard deviation of surface elevation within GOSAT ground pixel: <80m<br>Blended albedo: <0.9 | - |
| Consideration in scattering | 4 extinction profiles (two aerosol types , water and ice cloud) | logarithms of the mass mixing ratios of fine-mode aerosols and coarse mode aerosols with aerosol optical properties based on SPRINTARS V3.84 | Aerosol profile scaling of 2 different aerosol types; cloud extinction profile scaling | Aerosol particle number concentration, aerosol size parameter, aerosol height | - |
| Bias correction | $X'_{CO2} = X_{CO2} - 0.5 - 0.155 * (\Delta P_{s,cld} + 2.7) + 10.6 * (\alpha'_3 - 0.204) + 0.0146 * (\Delta GRAD_{CO2} - 35) + 12.8 * (AOD_{DUST} - 0.01)$ | - | Via a regression analysis of the difference between GOSAT and TCCON $XCO_2$ land observations. See details in the product user guide | $X'_{CO2} = X_{CO2} * (1.002837 + 2.1176e - 5 * \phi)$<br>$\phi$: the aerosol filter | - |
| Sources | Osterman et al., 2016; O'Dell et al., 2012; D.Wunch et al., 2011b. | NIES (GOSAT Project Office), 2015; Yoshida et al., 2013; D.Wunch et al., 2011b. | Hew, 2016; GHG-CCI group at University of Leicester, 2014. | Detmers et al., 2015; Hasekamp et al., 2015 | Buchwitz et al. 2016. Reuter et al., 2013. |

*The first represents mean biases, and the second represents overall standard deviations.*



Within the study area, the total numbers of valid GOSAT XCO$_2$ observations are 3345, 3556, 2282, 3685 and 2796 in
ACOS, NIES, OCFP, SRFP and EMMA, respectively. Figure 2 shows the number of available XCO$_2$ retrievals during the 4
seasons (spring: MAM; summer: JJA; autumn: SON; winter: DJF). It can be seen that the number of available XCO$_2$
retrievals is clearly smaller in spring and summer than that in autumn and winter due to different meteorological conditions
and data-screening processes. The cloudiness in spring and summer caused by the monsoon climate disturbs satellite
observation, while the smaller data number in the west of 110 °E is due to frequent dust storm in the Taklimakan Desert.

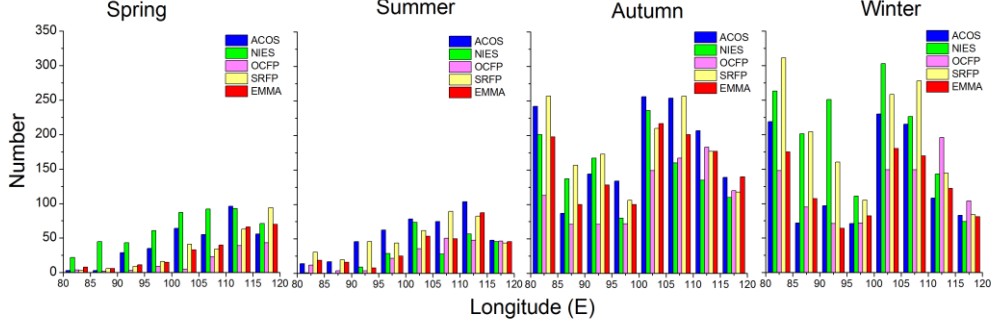


**Fig. 2. Number of single scans from five GOSAT-XCO$_2$ data sets over each 5x5 ° cells in spring, summer, autumn and winter from**
**March 2010 to February 2013. Spring: MAM; summer: JJA; autumn: SON; winter: DJF.**
**2.3 XCO$_2$ simulations from GEOS-Chem**
We use GEOS-Chem version 10-01 driven by GEOS-5 and the details of the main input emissions are as follows: 1)
Fossil fuel fluxes are taken from ODIAC version 2013. On the other hand, we also introduce the new emission data set
CHRED for the Chinese mainland. 2) The balanced biosphere CO2 uptake and emission fluxes are taken from the Simple
Biosphere Model version 3 (SiB3) [Messerschmidt et al. 2012]. 3) Biomass emissions are taken from Global Fire Emission
Database version 4 (GFEDv4) (Giglio et al., 2013). 4) Ocean fluxes are taken as Takahashi et al. (2009) suggested. The input
emissions for the GEOS-Chem CO$_2$ simulation are described in Nassar et al. (2010), although we have used some of the
most recent updates available in the GEOS-Chem version 10-01 and the Harvard–NASA Emission Component version 1.0
(HEMCO) module (Keller et al., 2014), a versatile component for emissions in atmospheric models. Higher model resolution
is very important in the calculation of the concentrations of atmospheric gases, especially over land where topography
smoothing (compared to reality) is determined by horizontal resolution (Ciais et al., 2010). Considering this, GEOS-Chem at
0.5 ° (latitude) x 0.666 ° (longitude) horizontal resolution, the nested grid model in China, was taken for the CO$_2$ simulation
with boundary conditions provided by the global model at 2 ° (latitude) x 2.5 ° (longitude) resolution. We made a restart file
with 386.4 ppm for both the global simulation and the nested simulation on 1 January 2009 based on NOAA ESRL data.
Both the global model and the nested-grid model were run twice, driven by the same CO$_2$ fluxes from January 2009 to
February 2013 except that the ODIAC was chosen for the first run and CHRED for the second as the input fossil-fuel fluxes
over the Chinese mainland. With an average for local hours between 12 pm and 13:30 pm, model CO$_2$ profiles were





presented from January 2010 to February 2013, allowing sufficient time for the high-resolution model to adjust to transients
introduced by the initialization of the model on 1 January 2009. The pressure-weighting function described in Connor (2008)
was applied to translate level-based modeling $CO_2$ to $XCO_2$. Figure 3 presents the spatial difference of emissions over the
Chinese mainland between CHRED and ODIAC at a horizontal resolution of 1°x1°. The values of emissions are mostly
larger in CHRED than in ODIAC, as shown in Fig. 3, and this difference tends to be large in the eastern part of our study
area. In addition, the difference in their total emissions, 10.38 Pg $CO_2$ (CHRED) to 9.64 Pg $CO_2$ (ODIAC), is not small.
ODIAC is also found to exhibit an overestimation of emissions in large cities compared to CHRED.

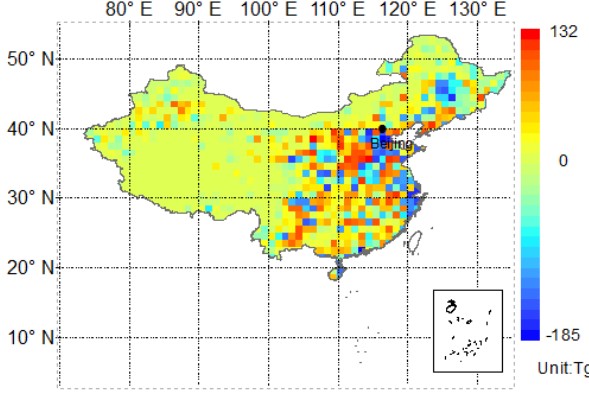


**Fig. 3. Difference of annual total anthropogenic $CO_2$ emissions between CHRED and ODIAC in 2012 in China, where the black**
**dot represents Beijing, the capital of China.**
For each 1°x1° grid, the corresponding annual $CO_2$ emissions in the years from 2009 to 2012 were allocated by the ratio
of emissions in CHRED to that in ODIAC in 2012. We acquired the new input inventory of $CO_2$ emissions, CHRED, by
scaling the obtained yearly emissions with the ratio of monthly emissions to the yearly ones in ODIAC. In this way, we have
altered the spatial and temporal distribution, but not at temporal scales finer than monthly. This is expected to be an
improvement upon the current ODIAC emission values.
The annually averaged $XCO_2$ driven by both CHRED and ODIAC are calculated and shown in Fig. 4. The impact of
emission deviations of CHRED from ODIAC is significant, with an average $XCO_2$ increase of 0.7 ppm over China. There
are also obvious differences in spatial patterns, especially in Northwest China, Northeast China, North China and South
China. Modeling $CO_2$ from CHRED increases values up to 0.7 ppm in most parts east of 100°E with a maximum at 1.4
ppm compared to that from ODIAC. The increase in the annual mean, which should not be ignored, is approximately 1 ppm
east of 110°E in the latitude zone of 37°N~ 42°N. Modeling the $XCO_2$ data set from CHRED is used to compare with
satellite-retrieved $XCO_2$ in our following experiment.





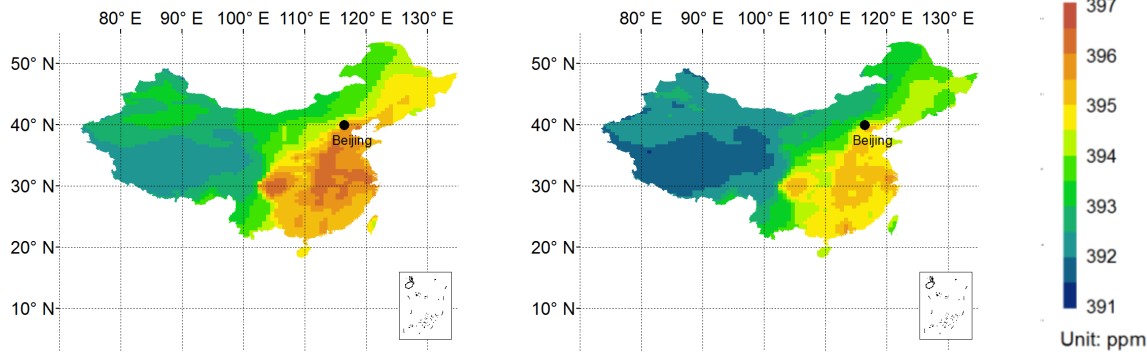


**Fig. 4. The annual mean XCO₂ concentrations driven by CHRED (left) and by ODIAC (right) in 2012 with GEOS-Chem in China,**
**where the black dot represents Beijing, the capital of China.**

**2.4 Aerosol optical depth and surface albedo data**

The monthly mean aerosol data was collected from the NASA Earth Observing System's Multi-angle Imaging Spectro-
radiometer (MISR) Level 3 Component Global Aerosol Product, downloaded from the website https://eosweb.larc.nasa.gov
/project/misr. The released GLASS (Glass Land Surface Satellites) albedo is used, which is a gapless, long-term continuous
and self-consistent data-set with accuracy similar to that of the Moderate Resolution Imaging Spectrometer (MODIS)
MCD43 product (Liu et al., 2013). Data were downloaded from the website http://glcf.umd.edu/data/abd/.

**3 Quantification of agreement of XCO₂ retrievals from five algorithms in the same footprints**

We focus on the difference of each footprint XCO₂ retrieval in this section. Comparison of XCO₂ from the five algorithms
with GEOS-Chem simulations driven by CHRED, and pairwise comparisons of XCO₂ between algorithms were calculated
as a quantified indicator of their differences.

**3.1 Comparisons with GEOS-Chem simulations**

We used the nested GEOS-Chem simulation XCO₂ as a baseline to quantify the regional consistency of the five algorithms.
Our output model CO₂ profile is the averaged concentration during the local hours 12:00-13:30 pm corresponding to the
local time of overpass and locations (latitude and longitude) of GOSAT. To compare XCO₂ retrievals from ACOS, NIES,
OCFP, SRFP and EMMA, corresponding GEOS-Chem XCO₂ data were created by applying averaging kernels from each
algorithm to model CO₂ profiles as suggested by Rodgers (2003). Correlation diagrams of XCO₂ between GEOS-Chem (X)
and GOSAT (Y) for the five algorithms are shown in Fig. 5. The regression slope (a), the coefficient of determination ($R^2$),
the correlation coefficient (r), and biases of GOSAT (Y) from GEOS-Chem(X) are also shown in the inset of each panel.
It can be found that the linear fits and the correlations with GEOS-Chem are better for ACOS, OCFP and EMMA ($R^2$
approximately 0.66) than for either NIES or SRFP ($R^2$ approximately 0.59). The regression slope is the closest to unity in the




OCFP panel (0.94), which means the best similarity in variation. The slope of GEOS-Chem vs ACOS ranks second (a=0.87)
while it is less than 0.8 vs NIES and SRFP. The bias relative to GEOS-Chem is within 0.5 ppm for ACOS, SRFP, and
EMMA, while it is 2 ppm and 1.2 ppm for NIES and OCFP, respectively.

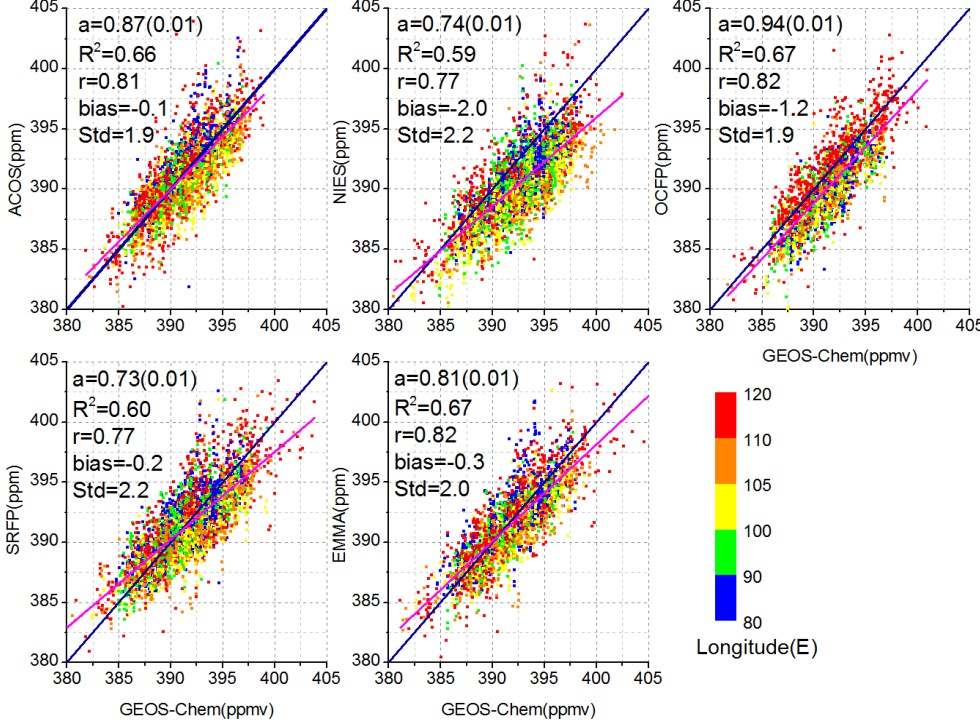


**Fig. 5: Correlation diagrams of GOSAT XCO$_2$ (Y) for the five algorithms versus GEOS-Chem (X) XCO$_2$ and linear fit statistics (insets of panels). GEOS-Chem data are selected corresponding to locations and time of XCO$_2$ from the five algorithms in cells. Deep blue solid lines represent a 1:1 line, and the magenta lines demonstrate the best fit for all observations. Colored points represent XCO$_2$ for different cells: blue-[80 °E, 90 °E], green-[90 °E, 100 °E], yellow-[100 °E, 105 °E], orange-[105 °E, 110 °E], and red-[110 °E, 120 °E] in the latitude zone [37 °N, 42 °N].**

217        Table 2 shows the biases and number of samples used between each algorithm and GEOS-Chem in each cell. It can be

seen that the biases relative to GEOS-Chem in all cells are below 1 ppm for ACOS, SRFP and EMMA, which implies better
consistency with GEOS-Chem regionally than NIES and OCFP. NIES presents values 1.2-3.1 ppm lower than GEOS-Chem
in all cells excluding the cell of 115 °E, which is because no corrections were implemented to reduce the existing systematic
biases in the NIES data set (Yoshida et al., 2013). The bias of OCFP relative to GEOS-Chem is larger than 1.2 ppm toward
the west of 110 °E, while it is 0.1 ppm toward the east of 110 °E. The standard deviations of the five algorithms with GEOS-
Chem range from 1.4 ppm to 2.7 ppm in all cells.





**Table 2. The biases relative to GEOS-Chem for five algorithms in each cell. The values in parentheses are the biases and their**
**standard deviations (upper values) and the number of samples (lower values) for each algorithm.**

| Left longitude of cells(°E) | 80 | 85 | 90 | 95 | 100 | 105 | 110 | 115 |
|---|---|---|---|---|---|---|---|---|
| ACOS | 0.7(1.6) | 0.5(1.6) | -0.4(1.4) | -0.3(1.5) | -0.7(1.7) | -0.7(1.7) | 0.0(2.2) | 0.5(2.1) |
| | 478 | 179 | 316 | 303 | 629 | 599 | 515 | 326 |
| NIES | **-1.4**(1.7) | **-1.6**(1.8) | **-1.6**(1.8) | **-2.3**(2.5) | **-3.0**(1.9) | **-3.1**(2.2) | **-1.6**(2.5) | -0.7(2.4) |
| | 487 | 383 | 470 | 281 | 700 | 506 | 428 | 301 |
| OCFP | **-1.8**(1.4) | **-1.8**(1.5) | **-2.2**(1.4) | **-1.2**(2.0) | **-2.3**(1.6) | **-1.5**(1.6) | -0.1(1.9) | -0.1(2.1) |
| | 277 | 172 | 149 | 175 | 339 | 390 | 466 | 314 |
| SRFP | 0.1(1.9) | 0.0(1.8) | 0.2(1.7) | -0.2(2.0) | **-1.2**(1.9) | -0.6(2.7) | 0.2(2.4) | 0.0(2.4) |
| | 602 | 387 | 388 | 271 | 571 | 659 | 467 | 340 |
| EMMA | 0.6(1.8) | 0.2(2.0) | -0.4(1.4) | -0.2(1.7) | -0.8(1.8) | -1.0(2.0) | -0.1(2.1) | -0.1(2.1) |
| | 400 | 229 | 211 | 222 | 484 | 460 | 453 | 337 |

**3.2 Pairwise comparisons of XCO$_2$ between algorithms**
We made comparisons of geometric and timely matching pairs XCO$_2$ between algorithms in each cell. The pairs of XCO$_2$
retrievals were matched between two algorithms timely in the same day, geometrically located within ±0.01 °in latitude and
longitude. Figure 6 shows pairwise comparisons of XCO$_2$ retrievals between two algorithms that demonstrate the regression
slope (a), the coefficient of determination (R$^2$), the correlation coefficient (r), the number of matching pairs (n) and the biases
between every pair of algorithms.
It can be seen from Fig. 6 that ACOS generally demonstrates the best agreement with other algorithms (top panel) and
the best agreement with EMMA with the greatest correlation of 0.95, a slope of 1.0 and bias of 0.3 ppm among all the
pairwise comparisons between algorithms. OCFP generally presents biases larger than 1.4 ppm with other algorithms except
for a value of 0.1 ppm compared to NIES. EMMA vs. ACOS and EMMA vs. SRFP present better agreement, with a
coefficient of determination greater than 0.87, as EMMA integrates products from seven individual algorithms [Reuter et al.,
2013], and the fractions in our study area are 36.7%, 30% and less than 18% from SRFP, ACOS and others, respectively.
It can also be seen from the colored points in Fig. 6 that matching pairs of XCO$_2$ for OCFP versus ACOS, SRFP and
EMMA mostly concentrated along the 1:1 line in the eastern cells of 105-120 °E (orange and red points) but drifted from the
1:1 line in the western cells of 80-100 °E (blue and green points).





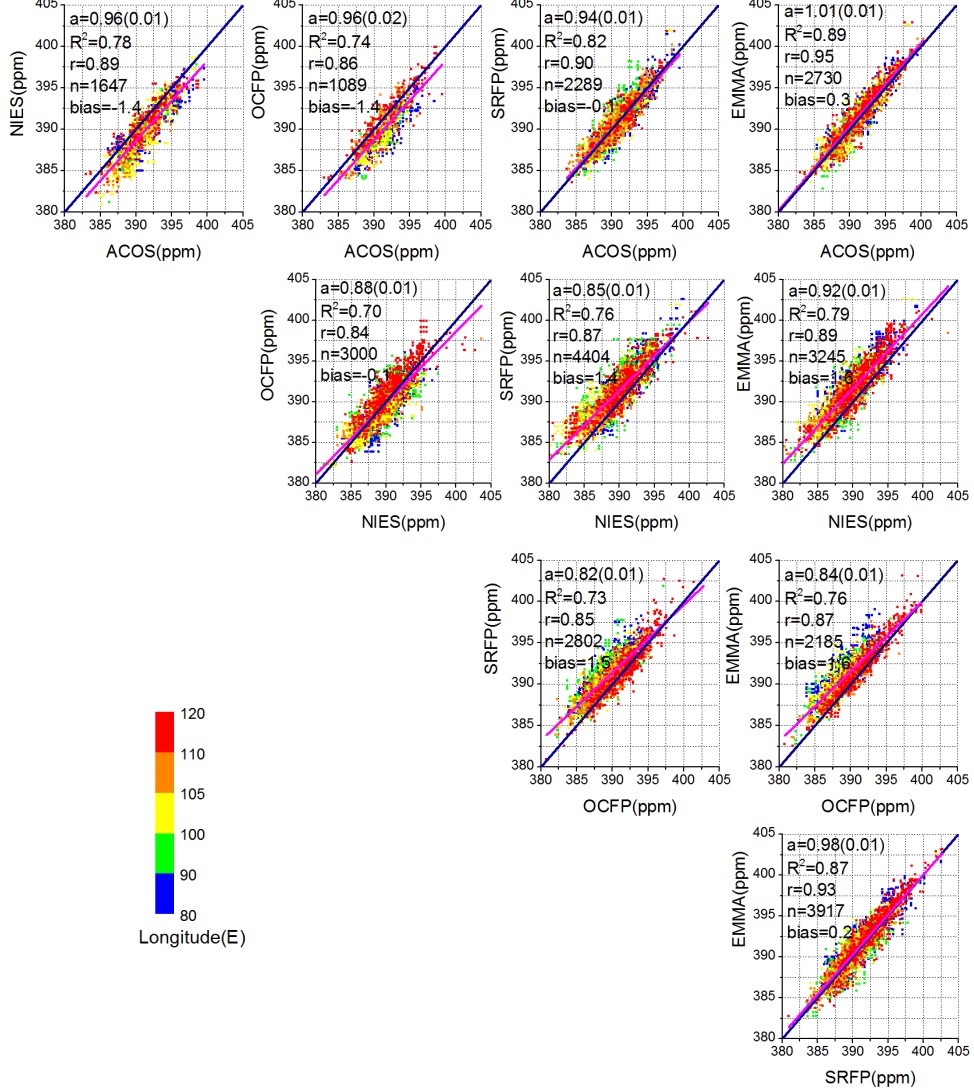


**Fig. 6: Algorithm correlation diagrams and statistical characteristics (insets of panels). GOSAT-Y observations were selected over land within ±0.01 °latitude/longitude of each GOSAT-X observation and in the same day. Deep blue solid lines represent a 1:1 line, and the magenta ones display the best fit for all observations. Colored points represent XCO$_2$ for different cells: blue-[80 °E, 90 °E], green-[90 °E, 100 °E], yellow-[100 °E, 105 °E], orange-[105 °E, 110 °E], and red-[110 °E, 120 °E] in the latitude zone [37 °N, 42 °N].**

The differences(biases) of matching pairs (the number ranging from 11 to 945) of XCO$_2$ between two algorithms, moreover, were calculated for each cell as shown in Table 3, and the totally averaged absolute differences of matching pairs of XCO$_2$ for an algorithm with the other algorithms were also calculated in each cell as shown in Table 4.

It can be found from Table 3 that the difference is mostly less than 1 ppm in those eastern cells with a longitude greater than 105 °E, and their consistency can be seen in Fig. 6 (red points between 110-120 °E) as well. The differences that are larger than 2 ppm are located in western cells with longitudes less than 105 °E, and these differences are mostly shown in OCFP vs. other algorithms. The total differences shown in Table 4, moreover, indicate that the differences of the five



algorithms tend to be similar to the results of matching pairs of $XCO_2$ (Table 3), and OCFP presents the largest difference up
to 2 ppm in the western cells of 80-90 °E.
**Table 3. Differences (ppm) between two algorithms (column algorithm minus row algorithm) for each cell, where values in**
**parentheses are the corresponding standard deviations.**

| | * | NIES | OCFP | SRFP | EMMA | * | NIES | OCFP | SRFP | EMMA |
|---|---|---|---|---|---|---|---|---|---|---|
| ACOS | | -1.4(1.2) | **-2.6**(1.2) | -0.5(1.2) | 0.2(1.0) | | **-1.6**(1.6) | **-2.0**(1.1) | -0.2(1.2) | 0.2(1.1) |
| NIES | 80 | | -0.9(1.4) | 1.1(1.4) | **1.7**(1.5) | 100 | | -0.4(1.4) | 1.4(1.5) | **1.6**(1.4) |
| OCFP | °E | | | **2.0**(1.2) | **2.6**(1.5) | °E | | | **1.7**(1.3) | **1.9**(1.4) |
| SRFP | | | | | 0.4(1.1) | | | | | 0.3(1.1) |
| ACOS | | **-2.0**(1.3) | **-1.9**(1.2) | -0.1(1.2) | 0.5(0.9) | | **-1.6**(1.3) | -0.6(1.4) | 0.2(1.2) | 0.2(0.9) |
| NIES | 85 | | -0.4(1.6) | 1.5(1.3) | **2.0**(1.5) | 105 | | 0.2(1.5) | 1.2(1.3) | 1.5(1.3) |
| OCFP | °E | | | **2.3**(1.4) | **2.7**(1.5) | °E | | | 1.0(1.3) | 1.0(1.0) |
| SRFP | | | | | 0.2(1.2) | | | | | 0.2(0.9) |
| ACOS | | -1.2(1.1) | **-1.7**(1.1) | 0.8(1.4) | 0.5(0.8) | | -1.2(1.3) | -0.9(1.4) | 0.0(1.4) | 0.4(1.1) |
| NIES | 90 | | -0.8(1.4) | **2.0**(1.4) | 1.5(1.2) | 110 | | 0.7(1.3) | 1.5(1.6) | 1.5(1.3) |
| OCFP | °E | | | **2.4**(1.5) | **2.0**(1.3) | °E | | | 0.5(1.2) | 0.7(1.0) |
| SRFP | | | | | -0.1(1.1) | | | | | 0.0(1.3) |
| ACOS | | **-3.0**(1.1) | -0.9(1.7) | -0.3(1.2) | 0.0(1.1) | | -0.6(1.3) | 0.1(1.0) | -0.1(1.0) | 0.5(1.0) |
| NIES | 95 | | 0.5(2.1) | 1.3(2.0) | **1.7**(1.9) | 115 | | 0.8(1.5) | 0.9(1.3) | 1.3(1.5) |
| OCFP | °E | | | **1.8**(1.6) | 1.4(1.1) | °E | | | 0.2(1.3) | 0.5(1.0) |
| SRFP | | | | | 0.2(1.3) | | | | | 0.6(0.9) |

**The columns labeled with * represent the left longitude of cells (°E).**
**Table 4. The average (ppm) of the absolute differences of the target algorithm (in column) matching all other algorithms for each**
**cell. Values in parentheses are the corresponding standard deviations.**

| Left longitude of cells(°E) | 80 | 85 | 90 | 95 | 100 | 105 | 110 | 115 |
|---|---|---|---|---|---|---|---|---|
| ACOS | 1.5(0.8) | 1.4(0.7) | 1.2(0.4) | 1.6(1.0) | 1.4(0.6) | 1.1(0.4) | 1.1(0.2) | 0.9(0.2) |
| NIES | 1.6(0.2) | 1.8(0.4) | 1.6(0.4) | **2.2**(0.6) | 1.6(0.3) | 1.5(0.3) | 1.5(0.3) | 1.3(0.2) |
| OCFP | **2.2**(0.6) | **2.1**(0.6) | 1.9(0.5) | 1.7(0.2) | 1.7(0.4) | 1.2(0.1) | 1.1(0.1) | 1.0(0.2) |
| SRFP | 1.3(0.5) | 1.4(0.7) | 1.6(0.8) | 1.4(0.6) | 1.3(0.5) | 1.1(0.3) | 1.2(0.4) | 1.0(0.2) |
| EMMA | 1.6(0.9) | 1.6(1.0) | 1.3(0.6) | 1.3(0.6) | 1.3(0.6) | 1.1(0.5) | 1.1(0.4) | 1.0(0.4) |


To summarize the quantification and analysis in this section, $XCO_2$ retrievals from three algorithms, ACOS, EMMA and
SRFP are mostly consistent, and the bias of ACOS relative to GEOS-Chem is the least among the five algorithms. The
difference of $XCO_2$ from cross-comparing five algorithms likely tends to be less in cells east of 100°E than that in the cells
west of 100°E.





## 4 Comparison of the spatio-temporal pattern revealed by $XCO_2$ from the five algorithms and simulation

We used a combination of sine and cosine trigonometric functions to statistically fit the seasonal variation of $XCO_2$, which was originally proposed by Keeling et al. (1976) and has been applied extensively in many studies (Thoning et al. 1989; Kulawik et al., 2016; Lindqvist et al., 2015; Zeng et al., 2016; He et al., 2017). Better attributions are thus obtained for $XCO_2$ variation in the seasonal cycle and in spatial background patterns by filtering the noise and filling gaps in the original $XCO_2$ data.

First, the monthly averaged $XCO_2$ was calculated in each cell using $XCO_2$ retrievals; then, the fit function (Keeling, 1976), expressed as the following equation [1], was applied to the monthly averaged $XCO_2$ from March, 2010 to February, 2013 for the five algorithms and GEO-Chem.

$$X(t) = A_1 \sin 2\pi t + A_2 \cos 2\pi t + A_3 \sin 4\pi t + A_4 \cos 4\pi t + A_5 + A_6 t \quad [1]$$

where t represents elapsed time in years, $A_1$-$A_4$ are the coefficients determining the seasonal cycle, $A_5$ represents the initial state of $XCO_2$ with seasonal variation removed, which can be regarded as the corresponding background concentration, and $A_6$ is the slope of the linear part in the yearly increase ignoring the minor non-linear part. To derive $A_1$-$A_6$ with the above formula, least squares were applied to fit the input monthly weighted means with the corresponding standard deviations as measures of errors. The monthly weighted means (e.g., X (t)) and the corresponding standard deviations in each cell were calculated with the weights inversely proportional to the square of retrieval uncertainty in each observation point.

The accuracy of fitting X(t) depends on the number of gaps in the available $XCO_2$ retrievals in time and in space resulting from the filtering mechanism for quality controlling. We introduce the Pearson's correlation, hereafter referred to as R, between the input and the predicted results from equation [1] and the unit weight mean square error hereafter referred to as σ, in fitting as an uncertainty to judge whether the fitting results are reasonable or not. In addition, we applied equation [1] to the GEOS-Chem dataset. Since atmospheric transport models do not share the same error sources with satellite-retrieved algorithms and without data gaps, GEOS-Chem provides helpful a priori information for reference.

### 4.1 Seasonal variation of $XCO_2$ retrievals

The time series in each cell are acquired for each algorithm with the above formula [1]. The monthly fitted $XCO_2$ from March 2010 to February 2013 in each cell for five algorithms as well as GEOS-Chem is shown in Fig. 7. The seasonal amplitudes (the difference between seasonal cycle maximums and minimums) and uncertainty of the fitting function as described by R and σ above are demonstrated in Table 5.




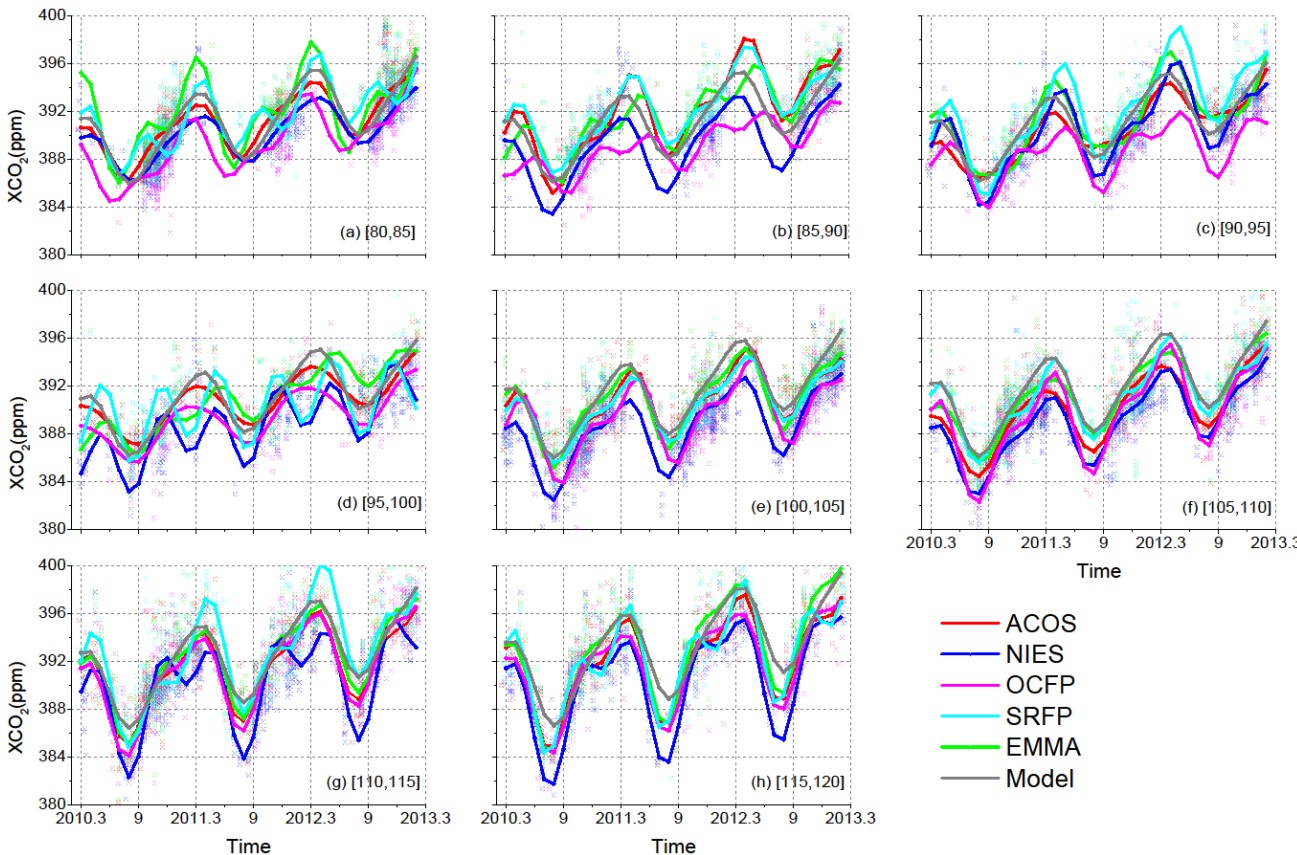

295

**Fig. 7: The time series from March 2010 to February 2013 in eight cells from the western cell of (a) to the eastern end cell of (h), where colored lines represent the fitting seasonal change trend of the five GOSAT-XCO₂ datasets from five algorithms, and the colored points represent single XCO₂ retrievals corresponding to five algorithms according to line color: red is for ACOS, blue for NIES, magenta for OCFP, cyan for SRFP and green for EMMA. The gray line is the fitting seasonal change trend of XCO₂ simulated by GEOS-Chem.**





**Table 5: Results of fitted seasonal cycle trend and uncertainty of fitting results in each cell for five algorithms and GEOS-Chem, The symbols "–" means that filtered results are not available due to large uncertainty judged by R and σ**

| Left longitude of cells (°E) | 80 | 85 | 90 | 95 | 100 | 105 | 110 | 115 |
|---|---|---|---|---|---|---|---|---|
| Seasonal cycle amplitude (ppm) | | | | | | | | |
| ACOSv3.5 | 5.1 | 7.8 | 3.7 | 4.0 | 6.6 | 5.9 | 8.0 | 9.3 |
| NIES | 4.3 | 6.9 | 7.8 | - | 7.1 | 6.4 | 9.5 | 10.7 |
| OCFP | 5.3 | 3.5 | - | 3.9 | 7.7 | 9.2 | 8.4 | 8.6 |
| SRFP | 6.3 | 6.5 | 8.9 | - | 5.9 | 7.4 | 10.4 | 10.7 |
| EMMA | - | - | 6.6 | - | 7.3 | 5.4 | 8.1 | 10.1 |
| GEOS-Chem | 6.3 | 5.9 | 5.7 | 5.6 | 6.5 | 6.9 | 7.2 | 7.9 |
| σ(Unit weight mean square error in fitting)(ppm) | | | | | | | | |
| ACOSv3.5 | 1.2 | 1.6 | 1.6 | 0.6 | 1.1 | 1.2 | 0.4 | 1.0 |
| NIES | 0.7 | 1.1 | 1.0 | 3.0 | 1.1 | 1.1 | 1.5 | 1.3 |
| OCFP | 0.7 | 0.9 | 1.5 | 1.4 | 1.9 | 1.1 | 0.8 | 0.9 |
| SRFP | 1.6 | 0.7 | 1.3 | 3.3 | 0.8 | 0.8 | 1.0 | 1.0 |
| EMMA | 2.7 | 1.5 | 1.1 | 2.1 | 1.2 | 0.9 | 0.6 | 0.6 |
| GEOS-Chem | 0.1 | 0.1 | 0.1 | 0.1 | 0.1 | 0.1 | 0.1 | 0.1 |
| R ( Correlations between fitted XCO2 and monthly averaged original XCO2 in each cell) | | | | | | | | |
| ACOSv3.5 | 0.92 | 0.92 | 0.91 | 0.95 | 0.91 | 0.91 | 0.98 | 0.94 |
| NIES | 0.89 | 0.91 | 0.94 | 0.68 | 0.96 | 0.95 | 0.89 | 0.92 |
| OCFP | 0.90 | 0.84 | 0.79 | 0.84 | 0.93 | 0.93 | 0.93 | 0.96 |
| SRFP | 0.83 | 0.94 | 0.92 | 0.40 | 0.95 | 0.94 | 0.93 | 0.90 |
| EMMA | 0.84 | 0.75 | 0.86 | 0.78 | 0.93 | 0.93 | 0.97 | 0.97 |
| GEOS-Chem | 1.00 | 1.00 | 0.99 | 0.99 | 0.99 | 0.99 | 0.99 | 0.99 |

Viewing the attribution of $XCO_2$ in each cell from Fig. 7 and Table 5, we can generally find that the seasonal variations from all $XCO_2$ retrievals show similar changing trends, except for one extra seasonal cycle maximum being misidentified in some cases mainly due to weaker data constraints for fitting. The timely changing patterns (seasonal cycle phases) of all algorithms demonstrate better agreement in the eastern four cells from 100°E to 115°E than those in the western four cells from 80°E to 95°E.. The correlation coefficients of fitting $XCO_2$ in Table 5 are also significantly greater in the eastern four cells than those in the western four cells. As a result, the longitude 100°E tends to be a regional border presenting better consistency of $XCO_2$ among the five algorithms in its eastern cells than those in western cells.





Comparing the five algorithms with GEOS-Chem, one specific result is presented in the eastern-most two cells from
110°E to 120°E, in which the seasonal amplitudes of $XCO_2$ are significantly higher from the five algorithms while the
magnitudes of $XCO_2$ in summer are lower than those from GEOS-Chem as shown in Table 5 and Fig. 7. There is strong CO2
absorption from farming activities of wheat and corn in the summer (Lei et al., 2010) and anthropogenic $CO_2$ emission from
extra winter heating in these eastern cells. This result is in agreement with an investigation of results in the whole Chinese
mainland (Lei et al., 2014) and at 120-180°E over the globe (Lindqvist et al., 2015), which is likely due to the
underestimated widespread bio-ecological $CO_2$ uptake changes that occurred over the past 50 years in atmospheric transport
models (Graven et al., 2013).
The $XCO_2$ values from NIES (blue in Fig. 7) are overall lower than those from the other algorithms, which is due to the
uncorrected systematic errors -1.2 ppm (refer to Table 1). The seasonal variations from OCFP (magenta in Fig. 7) are of the
overall seasonal changing trend of $XCO_2$ in cells west of 100°E. The seasonal amplitudes of OCFP presented in Table 5,
moreover, are abnormally the lowest in a cell (85-90°E) and the highest in a cell (105-110°E). SRFP and NIES show two
abnormal peaks in a cycle of a year in the cell of 95 °E, while some great values of σ and small values of R, shown in bold in
Table 5, indicate poor fitting mostly in the same cell (95-110°E). These results are likely induced by large gaps in the
available $XCO_2$ data in time series, which produces a poor fitting constraint.

## 4.2 Spatio-temporal pattern of detrended $XCO_2$

We calculated the seasonal averages of the $XCO_2$ background concentration in each cell after removing the linear yearly
increase using the fitting time series of $XCO_2$ for the five algorithms and GEOS-Chem. The spatio-temporal continuous
pattern of background $XCO_2$ was mapped by Linearly Interpolate Triangulation (Watson et al., 1984) using the seasonal
averages of $XCO_2$ background concentration in each cell for five algorithms and GEOS-Chem, as shown in Fig. 8 (on the
left). The spatio-temporal patterns of the differences in $XCO_2$ between the five algorithms and GEOS-Chem were mapped
and are also shown in Fig. 8 (on the right).






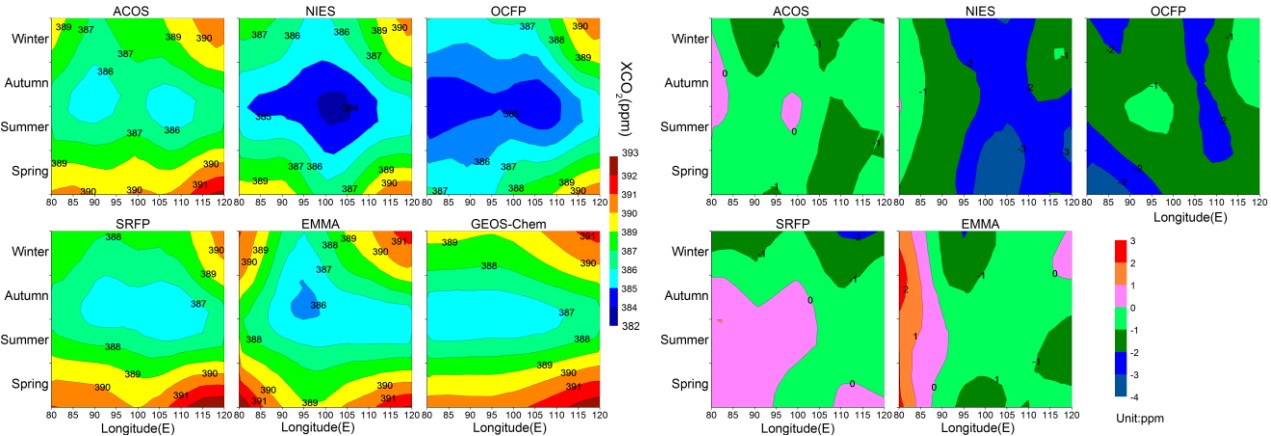


**Fig. 8: The spatial (in the study latitude band) and temporal (in seasons) changing patterns of detrended $XCO_2$ from ACOS, EMMA, NIES, OCFP, SRFP and GEOS-Chem (left) and the differences of detrended $XCO_2$ between ACOS, EMMA, NIES, OCFP and SRFP and GEOS-Chem.**

It can be seen from Fig. 8 (on the left) that the spatio-temporal patterns from the four algorithms of ACOS, EMMA, NIES and SRFP are generally the same, with an increase spreading outward from the center of each diagram and with the lowest $XCO_2$ located approximately 95°E-105°E and in the period of summer-autumn; meanwhile, OCFP and GEOS-Chem show a similar spatio-temporal pattern where the lowest value is not the center. Two common characteristics of $XCO_2$ spatio-temporal changes from five algorithms and GEOS-Chem can also be found: (1) the seasonal changes of $XCO_2$ are the same in any spatial cells, with lower $XCO_2$ in summer and autumn than that in spring and winter; and (2) spatial changes of $XCO_2$ generally demonstrate larger $XCO_2$ in the eastern cells than those in the western cells in any season.

Compared to those of GEOS-Chem, the spatio-temporal differences of ACOS and SRFP generally demonstrate the smallest values mostly ranging from -1 ppm to 1 ppm. $XCO_2$ values from both NIES and OCFP are lower than GEOS-Chem in space and time, while the $XCO_2$ difference from GEOS-Chem is mostly 1-3 ppm for NIES and 2 ppm for OCFP. As a combination of products including the other four algorithms, EMMA demonstrates the largest difference with GEOS-Chem in most of the western cells in all four seasons where the difference is mostly less than 1 ppm.

To summarize the quantification and analysis in this section, the spatio-temporal pattern of ACOS tends to be inconsistent with SRFP. Figure 8 shows two common characteristics among ACOS, NIES, SRFP and EMMA: (1) $XCO_2$ is lower in summer and autumn but higher in spring and winter. (2) $XCO_2$ is higher west of 90°E and east of 110°E, while it is lower in cells 90°E-110°E. In addition, $XCO_2$ values from NIES and OCFP are lower than those from other algorithms, especially in summer and autumn. A similarly high level is captured by ACOS, EMMA, NIES and SRFP generally in the western deserts with lower $CO_2$ emissions compared to the east, which has abundant emissions. This is distinct from ACOS and EMMA, while OCFP and GEOS-Chem both show an increasing trend from west to east in any season.





## 5 Discussion

In this section, an investigation was made into the most likely attribution of regional inconsistency, i.e., aerosols and albedo, and an additional evaluation was made of the latest released ACOS V7.3, the newer version of ACOS data retrieved by the OCO-2 algorithm.

### 5.1 Discussion of albedo and aerosol effects for XCO$_2$ retrieval

The above quantification and analyses indicate that generally good agreements are achieved among the five data sets in the eastern cells, while four out of five GOSAT-XCO$_2$ data sets present abnormal high concentrations in the western cells. It has been known that aerosols are the most important factor inducing errors in satellite-retrieved XCO$_2$ (Guerlet et al., 2013; Oshchepkov et al., 2013; Yoshida et al., 2013; O'Dell et al., 2012), while Aerosol Optical Depth (AOD) is greatly affected by high surface albedo because of the optical lengthening effect. For that reason, we investigate the spatial and temporal characteristics of aerosols and albedo in our study latitude band to probe the reason why high inconsistency of XCO$_2$ retrieval algorithms appears in western cells rather than in eastern cells with intensive human activities.

We collected MISR aerosol products (AOD at 555 nm) and GLASS albedo products. The spatial and temporal characteristics of albedo and AOD with seasons in the study area are revealed as shown in Fig. 9, in which they are mapped by the same method as Fig. 8. The seasonal mean AOD and albedo were calculated in spring (MAM), summer (JJA), autumn (SON), and winter (DJF) using the monthly mean AOD and black sky shortwave albedo from January 2010 to December 2012 for every cell.

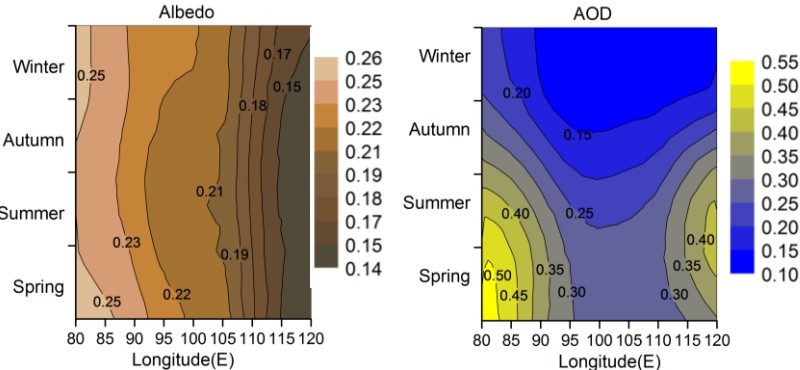

**Fig. 9: The temporal and spatial patterns of black sky short wave albedo (left) and aerosol optical depth at 555 nm (right). Colors represent albedo (left) and AOD (right).**

Albedo shows little temporal variation with a decreasing trend from west to east as shown in Fig. 9. In contrast with albedo, AOD follows a clear seasonal pattern of a higher level in spring and summer than in autumn and winter. The uplift of AOD in spring and summer is due to the higher frequency of Asian sand and dust storms for cells west of 105 °E. The main contributors to aerosol loading east of 110 °E are emissions from urban fugitive dust/fly ash, dust plumes from deserts in the western and northern China such as the Taklimakan deserts, industrial activities and residential heating (Zhang et al., 2012).





The inconsistency of XCO₂ from the five algorithms tends to be higher in spring and summer than in autumn and winter
in the Taklimakan Deserts in western cells, which is likely the combined effect of high aerosol and high brightness surface
(high albedo) on retrieval uncertainty.
From the above quantification and analysis, the pairwise differences between OCFP and other algorithms are 1 ppm
higher west of 105 °E than east of that, with a difference of 1.6 ppm over the whole study area. The obvious regional
characteristic probably relates to the assumption of a cirrus profile according to latitude (GHG-CCI group at University of
Leicester, 2014), which is unlikely to be reasonable in our study area. There exists a large amount of high clouds over the
Tibetan Plateau (Chen et al., 2005), which is located south of the study cells of 80 °E to 105 °E. The humidity and
atmospheric structure are mainly affected by the Tibetan Plateau, and there is a large difference in the cirrus profile between
the western cells and the eastern cells over our study area (Wang et al., 2012), which indicates that a uniform profile by
latitude will inevitably introduce errors.
The pairwise difference between NIES and other algorithms is 1.6 ppm on average, which is distinct among the
algorithms. Considering the complicated geographic environment in the study area, this distinct difference is likely related to
the presumptions of NIES in aerosol profiles and properties from an aerosol transport model (Table 1), by which cirrus
clouds are ignored and little information from observations is used in the retrieving process. Values from EMMA are found
to be abnormally large in spring and winter in the western cells from the spatio-temporal patterns (Fig. 8), which is not
exactly the same as with the other algorithms. Since data in EMMA are a combination of retrievals from multiple algorithms,
it may indicate the uncertainty in all algorithms under the circumstance of high albedo and AOD.
With the satellite-observed spectrum used for water and clouds, ACOS sets the initial aerosol types and AOD based on
a priori information. On the other hand, SRFP handles aerosol based on the property of number, size and height. Both of the
above two mechanisms function well since ACOS and SRFP are generally demonstrated to provide relatively better
performance.
Noticing that all algorithms differ in simulating scattering in the atmosphere, such as in the aerosol models, the
influence of scattering on retrieved XCO₂ is too significant to be ignored. Since it is possible for products from different
algorithms to agree with each other, there is no denying that satellite XCO₂ retrievals have the potential to provide more
accurate XCO₂ data. Optimization in the handling of aerosol scattering will improve the precision and accuracy of satellite
XCO₂ retrievals.
**5.2 Additional evaluation of the latest released ACOS V7.3**
The ACOS/OCO-2 research team released the latest version of the ACOS data ACOS V7.3 during the implementation and
completion of this study. We add the cross-comparisons of this version of the data set and other data sets including GEOS-
Chem, ACOS V3.5, NIES V02.XX, OCFP 6.0, SRFP V2.3.7 and EMMA V2.1.c in this section. ACOS V7.3 was created by
applying the XCO₂ retrieval algorithms of OCO-2 to GOSAT. Within the algorithm code of ACOS V3.5, the OCO-2
algorithm generating ACOS V7.3 data makes some changes in parameter settings, such as the surface pressure a priori





constraint and cloud ice properties, and it updates the manners of data processing, for example, the bias corrections and
filtering mechanism. The available data points, a total of 1980, were shown from March 2010 to February 2013 in Fig. 10,
where different colors and symbols in each panel represent the left longitude of cells into which retrievals fall. In cells west
of 90 °E, there are a few data points showing abnormal concentrations as high as above 400.0 ppm, which is higher than that
of data points in the east, where there are strong anthropogenic $CO_2$ emissions.

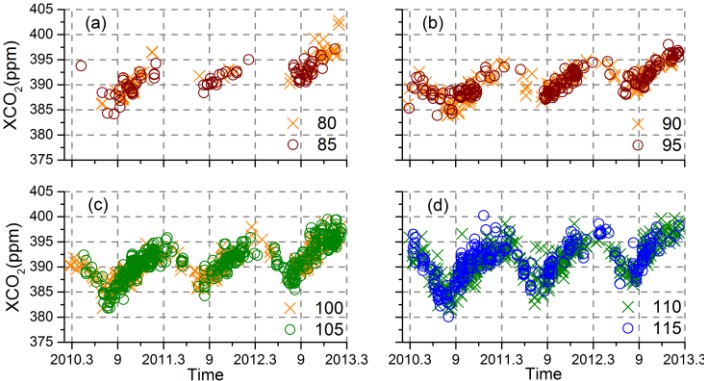


**Fig. 10.  The time series of data points from ACOS V7.3 during the period from March 2010 to February 2013. Different symbols**
**in each panel represent the left longitude of the cell into which a data point falls.**
We made cross-comparisons between ACOS V7.3 and other data sets. No bias was found in ACOS V7.3 from GEOS-Chem
with a standard deviation of 1.6 ppm and $R^2$=0.77. The comparison results in the cells are shown in Table 6. Generally,
ACOS V7.3 is in good agreement with all of them, which is reflected by correlation coefficients r that are above 0.85 and
greater than others, as shown in Table 6. The biggest differences up to 3.0 ppm for ACOS V7.3 are found from NIES and
OCFP in deserts cells, whereas differences from SRFP and EMMA are mostly within 1.0 ppm. This is similar to ACOS V3.5.
The total absolute difference from other algorithms (not including ACOS V3.5) is within 1.0 ppm in cells east of 110 °E but
above 2.0 ppm in cells west of 90 °E. It can also be found from Table 6 that the bias of ACOS V7.3 relative to GEOS-Chem
is within 0.3 ppm but above 1.3 ppm, in cells east and west of 90 °E, respectively.
Compared to the previous version, ACOS V3.5, ACOS V7.3 increases the average by approximately 0.2 ppm. In
comparison with the difference patterns with ACOS V3.5, the averages of the absolute differences between ACOS V7.3 and
the other four algorithms are similar (<0.1 ppm) and increase by an average of 0.6 ppm (2.1 ppm vs. 1.5 ppm) in cells east of
110 °E and west of 90 °E, respectively, while the biases relative to GEOS-Chem decrease approximately 0.3 ppm and increase
approximately 0.9 ppm in cells east and west of 90 °E, respectivley.
The comparison results further demonstrate inconsistency of $XCO_2$ among different datasets in the desert cells.




**Table 6. Differences between ACOS V7.3 and others (including GEOS-Chem and five other algorithms including ACOS V3.5, NIES, OCFP, SRFP and EMMA) in each cell (subtraction from ACOS V7.3). Values in parentheses are the corresponding standard deviations.**

| Left longitude of cells(°E) | 80 | 85 | 90 | 95 | 100 | 105 | 110 | 115 | r |
|---|---|---|---|---|---|---|---|---|---|
| GEOS-Chem | **-1.7**(1.5) 64 | -1.3(1.3) 85 | 0.1(1.2) 167 | 0.1(1.2) 191 | -0.1(1.3) 294 | 0.3(1.6) 448 | 0(1.7) 487 | 0(1.6) 244 | 0.88 |
| ACOS V3.5 | -0.4(0.9) 103 | -0.1(1.0) 48 | -0.1(1.0) 133 | -0.2(1.0) 189 | 0.0(1.1) 350 | -0.5(1.1) 391 | 0.2(1.2) 244 | -0.1(1.1) 126 | 0.93 |
| NIES | **-3.2**(1.2) 61 | **-1.9**(1.5) 100 | **-1.6**(1.2) 251 | -1.2(1.9) 123 | **-1.9**(1.4) 541 | **-1.8**(1.5) 317 | -1.2(1.6) 397 | -0.7(1.5) 277 | 0.87 |
| OCFP | **-3.1**(1.0) 66 | **-3.4**(0.9) 41 | **-2.2**(1.1) 157 | **-2.5**(1.5) 114 | **-2.1**(1.2) 297 | -1.5(1.1) 329 | -0.5(1.1) 396 | -0.1(1.0) 202 | 0.86 |
| SRFP | -0.8(1.3) 138 | -0.7(1.4) 145 | 0.3(1.3) 345 | -0.6(1.3) 337 | -0.4(1.3) 466 | -0.5(1.4) 631 | 0.3(1.4) 447 | 0.1(1.2) 247 | 0.89 |
| EMMA | -0.3(1.3) 113 | -0.5(1.4) 90 | 0.0(1.0) 190 | -0.4(1.4) 241 | -0.2(1.3) 405 | -0.3(1.2) 383 | 0.3(1.1) 390 | 0.5(1.1) 233 | 0.91 |
| Average absolute difference[1] for four algorithms above | 2.2(1.1) | 2.0(1.0) | 1.4(0.7) | 1.7(0.7) | 1.6(0.6) | 1.4(0.4) | 1.1(0.3) | 1.0(0.2) | |

*[1] **represents the average of absolute differences of ACOS V7.3 matching other algorithms including NIES, OCFP, SRFP and**
**EMMA for each cell.**

## 6 Conclusion

Although TCCON has been widely accepted as the standard for validation of satellite-based $XCO_2$ data, it is necessary to better understand the performance of $XCO_2$ in spatial and timely variations at a regional scale and especially for those regions where ground-based measurements of $XCO_2$ are not available, such as for the TCCON stations in China. We implement the quantification and assessment of the agreement of multiple algorithms for typical regions with various land covers and enhancement of anthropogenic $CO_2$ emissions including the megacity of Beijing from 80°E to 120°E in the same latitude band of 40°N to get better knowledge of the regional uncertainty and performance of GOSAT $XCO_2$ retrievals in China. Regional performance of $XCO_2$ products from six algorithms (ACOS, NIES, OCFP, SRFP, EMMA, OCO-2) as well as GEOS-Chem simulated $XCO_2$ are probed to obtain the regional uncertainty and attributions of GOSAT $XCO_2$ retrievals. In particular, we apply simulated $XCO_2$ at a high spatial resolution of 0.5° (latitude) x 0.666° (longitude) for a nested grid obtained by GEOS-Chem to assess the regional uncertainty of $XCO_2$ derived from satellite observations in China. In connection with the inconsistency of algorithms in eight cells, the characteristics of aerosol and albedo are investigated to discuss the further attribution of regional inconsistency of algorithms.



Summarizing the performance of five algorithms (ACOS, NIES, OCFP, SRFP and EMMA) in each cell based on the
above quantification and analysis from comparisons with GEOS-Chem, pairwise differences between algorithms and
agreement in time series among algorithms, we can obtain the following results in general: (1)The consistency among
algorithms is better in the east than in the west as the absolute difference from pairwise comparisons presents values of 0.9-
1.5 ppm in eastern cells covered by grassland, cropland and built-up areas and values of 1.2-2.2 ppm in western cells covered
by desert with a high-brightness surface;(2)ACOS and SRFP are more satisfying in characterizing spatio-temporal
patterns than other algorithms. To conclude, Table 7 presents the regional characteristics and a summary of the results above.
**Table 7. Summaries of our analyses above, including uncertainty, emissions, albedo, aerosol optical depth, regional differences in**
**footprint retrievals compared to GEOS-Chem, differences in footprint retrievals and agreement in time series among algorithms,**
**general differences in footprint retrievals and agreement in detrended $XCO_2$ compared to GEOS-Chem for algorithms.**

| Left longitude of cells (°E) | 80 | 85 | 90 | 95 | 100 | 105 | 110 | 115 |
|---|---|---|---|---|---|---|---|---|
| $CO_2$ emissions (Tg/year)[1] | 20.1 (24.1) | 11.2 (7.8) | 1.2 (2.7) | 35.8 (20.7) | 57.1 (15.6) | 515.2 (199.0) | 801.3 (600.3) | 821.9 (893.3) |
| Surface type | High brightness desert | | | Gobi desert | | Grassland | Cropland and built-up | |
| Albedo | 0.24-**0.26** | 0.23-**0.26** | 0.22-0.24 | 0.19-0.21 | 0.21-0.22 | 0.20-0.21 | 0.15-0.17 | 0.14-0.16 |
| AOD[2] | 0.22-**0.53** | 0.16-0.42 | 0.12-0.32 | 0.10-0.29 | 0.12-0.28 | 0.12-0.28 | 0.10-0.32 | 0.10-0.37 |
| Regional Summary in pairwise differences between algorithms | Less Consistency (mean absolute differences 1.2-2.2 ppm) The difference of OCFP is the greatest with most of the other algorithms (1.7-2.2 ppm); next is NIES (1.6-2.2 ppm). | | | | | Good consistency (mean absolute differences 0.9-1.5 ppm) ACOS is relatively the least (0.9-1.1 ppm) | | |
| Regional Summary compared to GEOS-Chem | Large biases, of which NIES is the greatest (1.4-3.1 ppm) and next is OCFP (1.2-2.2 ppm) | | | | | lesser biases (0.0-0.5 ppm) excluding NIES | | |
| | Similar in seasonal amplitude; | | | | | Seasonal amplitude from GEOS-Chem is lower than all of satellite retrieval algorithms. | | |
| Regional pairwise comparisons of ACOS V7.3 | Greater biases are presented with OCFP (1.5-3.4 ppm) and NIES (1.2-3.2 ppm) | | | | | Lesser biases (0.0-0.5 ppm) excluding NIES | | |
| General differences compared to GEOS-Chem | ACOS presents lowest values (bias -0.1 ppm Std[3] 1.9 ppm), next is SRFP (bias -0.2 ppm Std 2.2 ppm) NIES presents the greatest (bias -2.0 ppm, Std 2.2 ppm). | | | | | | | |
| Spatio-temporal patterns of $XCO_2$ compared to GEOS-Chem | ACOS and SRFP are similar to GEOS-Chem. OCFP is in better agreement with GEOS-Chem but the bias is larger. | | | | | | | |

**[1] represents the total emissions of $CO_2$ from CHRED (values without parentheses) and from ODIAC (values in parentheses) in**
**each cell in 2012. [2] is the range of averaged seasonal aerosol optical depth over a year. [3] is the standard deviation.**



The results, indicating that the discrepancies among algorithms are the smallest in eastern cells, which are the strongest
anthropogenic emitting source regions in China, implies that the uncertainty of $XCO_2$ is likely low in this area, which will be
sufficiently rigorous for us to apply it to GOSAT $XCO_2$ in assessment of anthropogenic emissions. Moreover, it was likely
that uncertainty in satellite-retrieved $XCO_2$ is attributed to the combined effects of aerosol and albedo. The large uncertainty
of $XCO_2$ must be improved further, even though many algorithms have endeavored to minimize the effects of aerosol and
albedo. With the launch of OCO-2 in 2014 and GOSAT-2 scheduled for 2018, the prospect of a large amount of useful
retrieved $XCO_2$ products is promising. Since low regional $XCO_2$ biases are necessary for accurately estimating regional
carbon sources and sinks, regional uncertainty should be paid more attention in the future.
Acknowledgements:  This research was supported by the National Research Program on Global Changes and Adaptation:
"Big data on global changes: data sharing platform and recognition" (Grant No. 2016YFA0600303, 2016YFA0600304). We
are grateful for NIES products from NIES GOSAT Project, albedo data from Beijing Normal University and supports from
GEOS-Chem team. ACOS V3.5 and ACOS V7.3 were produced by the ACOS/OCO-2 project at the Jet Propulsion
Laboratory, California Institute of Technology, and obtained from the JPL website, co2.jpl.nasa.gov.We are grateful for
aerosol data from Aeronautics and Space Administration (NASA). The satellite XCO2 products OCFP, SRFP and EMMA
have been obtained from the ESA project GHG-CCI website (http://www.esa-ghg-cci.org/) and the data providers Univ.
Leicester (OCFP product), SRON & KIT (SRFP) and Univ. Bremen (EMMA) have granted permission to use these data for
peer-reviewed publications.

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
