# Peer review of "Regional uncertainty of GOSAT XCO2 retrievals in China"

_Atmospheric Measurement Techniques, 2017_

## Referee Comment (RC1) · Anonymous Referee #1 · 27 Nov 2017

The paper shows an inter-comparison between the 5 main CO2 retrieval algorithms from GOSAT over China and a comparison to an atmospheric transport model to study regional "uncertainties" in XCO2 retrievals under different conditions (ranging from anthropogenic emissions of CO2 and aerosols to desert conditions with no CO2 emissions but natural aerosols).

The paper is interesting to the CO2 remote sensing community although in the end it stays rather inconclusive. The reason is that there is no absolute reference for the true XCO2 in this study. The conclusions that are being drawn are based on (in-)consistency between different retrieval algorithms and comparison to the GEOS-CHEM model and are hence to large extend speculative. The discussion on the aerosol and albedo effect stays qualitative while a more quantitative analysis would be of interest here.

I suggest to revise the paper to include a more quantitative analysis of the effect of aerosols and albedo on differences in retrieved XCO2 between different algorithms. This analysis should show to what extend the differences between algorithms, and between retrieved and models XCO2, are correlated with AOD and surface albedo. When such an analysis is included I recommend publication of the manuscript in AMT.

Other points: - How accurate are the XCO2 values modeled by GEOS-CHEM? The paper would benefit from a demonstration of the capability of GEOS-CHEM, for example from a comparions with TCCON (albeit outside the study region). - EMMA should be excluded from the analysis in this paper as it is not a retrieval algorithm itself but is composed from the different algorithms that are also analyzed in the present study. In fact, each EMMA value is the XCO2 retrieved by one of the algorithms that is closest to the median value for a given grid box. By including it in this study it correlates algorithm to itself. - A proper reference should be made to EMMA as a tool to study consistency between different algorithms, like is being done in the present study. -Line 132 states: " The recommended bias corrections are applied to the collected XCO2 data from ACOS, OCFP and SRFP". What is meant here? The files for both products already contain bias corrected products. Have these been used? - Line 364 stated:" while Aerosol Optical Depth (AOD) is greatly affected by high surface albedo because of the optical lengthening effect.". What is meant here? AOD is not affected by surface albedo. - The additional analysis of the new ACOS V7.3 product is confusing. It should either be used in the full analysis or the discussion should be shortened by only stating to what extend the conclusions would be different if the ACOS V7.3 product would have been used. The more detailed analysis could be moved to an appendix.

---

## Referee Comment (RC2) · Anonymous Referee #2 · 13 Dec 2017

Major points : See the comments from the other reviewer : - EMMA should be left out as it is the combined product of all other retrieval products shown - Shorten the part on the new version of ACOS, or use only the new version data - Provide a more quantitative analysis of the effect of aerosols and albedo on the observed differences between different algorithms - Provide some clear evidence of performance of GEOS-Chem wrt total column XCO2

Minor : Textual suggestions :

p.2 line 46 : I think you should leave out TanSat in that particular sentence as that instrument has not yet contributed to a better understanding of . . . as far as I know.

p.3 line 85-86 : rephrase 'that trend . . .to east' because unclear what is meant

p.9 GLASS albedo is used. For which wavelength is this albedo ?

table 2. Add to the table caption : All biases > 1 ppm are underlined. Change 'the values in parentheses are the biases and their . . .' → 'the values are the biases and –in parentheses- their. . .'

Table 3 table caption. What are the underlined values ?

p.18 line 350 ('To summarize the quantification. . . SRFP') : I do not understand this sentence given the data.

Fig. 8 Figure caption 'and the differences of detrended . . .. and GEOS-Chem' should that be '. . . with GOES-Chem' ?

p.21 line 423/424 I do not understand the sentence 'No bias was found . . . R2=0.77' based on what I see in Table 6. Also it is not consistent with what is written in line 429/430.

p. 23, line 462 results above → results described above

---

## Author Comment (AC1) · 9 Jan 2018

**Referee #1**

Responses to Anonymous Referee #1 on the manuscript of "Regional uncertainty of GOSAT XCO2 retrievals in China: Quantification and attribution"

**Referee #1: general:**

**-The paper is interesting to the CO2 remote sensing community although in the end it stays rather inconclusive. The reason is that there is no absolute reference for the true XCO2 in this study. The conclusions that are being drawn are based on (in-) consistency between different retrieval algorithms and comparison to the GEOS-CHEM model and are hence to large extend speculative.**

For inconclusive problem as you point out, we revised our analysis results concluded in Table 7. In this study, we aim to reveal regional uncertainty of GOSAT XCO2 retrievals via comparison and evaluation of consistence of multi-algorithms for GOSAT observations, and probe the reason why performances of $XCO_2$ from multi- algorithms are different in same regions. Our results are expected to give a reliable and valuable reference for application of XCO2 data in detection of carbon source and sink at a regional scale, e.g. the result gotten by our analysis, the better consistence of XCO2 from four algorithms (ACOS, NIES, OCFP, SRFP) in Eastern China with large anthropogenic CO2 emissions, can promote us to detect the anthropogenic enhancement of CO2 concentrations using these XCO2 data with confidence, and the result, the existing problems in deserts likely influenced by albedo and AOD, is expected to get attentions and improvement.

**Table 1. Summaries of our analyses for uncertainty of XCO2 retrievals obtained by GOSAT via inter-comparison of multi-algorithms above, including characteristics of regional emissions, albedo, aerosol optical depth, and summary of differences between algorithms and bias compared to GEOS-Chem.**

| Characteristics of regions and summary of algorithms | | Cells from 80 °E to 115 °E within 37°N-42°N | | | | | | | |
|---|---|---|---|---|---|---|---|---|---|
| | Regions Left longitude ( °E) | 80 | 85 | 90 | 95 | 100 | 105 | 110 | 115 |
| Characteristics of regions | $CO_2$ emissions (Tg/year)*[1] | Low emissions (1.2-57.1) | | | | | High emissions (515.2-**821.9**) | | |
| | Property of aerosol (AOD)*[2] | Dust (0.22-**0.53**) | | Clear (0.10-0.28) | | | | Urban (0.10-0.37)) | |
| | Surface types (albedo) | Sand desert with high brightness (0.20-**0.26**) | | | Gobi and grassland (0.19-0.22) | | | Cropland and built-up (0.14-0.17) | |
| Summary of uncertainty | Consistency of algorithms (pairwise mean absolute differences) | Less Consistency ( 1.0-1.6 ppm) | | | | | | Good consistency (0.7-1.1 ppm) | |

| Bias compared to GEOS-Chem (bias range) | Large biases (1.2-3.1 ppm) | lesser biases excluding NIES (0.0-0.5 ppm) |
|---|---|---|
| General performance of algorithms in spatio-temporal patterns of $XCO_2$ compared to GEOS-Chem | ACOS presents the lowest bias (-0.1$\pm$1.9 ppm); SRFP is next ( -0.2$\pm$2.2 ppm) NIES presents the greatest -2.0$\pm$2.2 ppm) | |

[1] represents the total emissions of $CO_2$ from CHRED in each cell in 2012. [2] is the range of averaged seasonal aerosol
optical depth over a year.

**-The discussion on the aerosol and albedo effect stays qualitative while a more quantitative analysis would be of**
**interest here. I suggest to revise the paper to include a more quantitative analysis of the effect of aerosols and**
**albedo on differences in retrieved XCO2 between different algorithms. This analysis should show to what**
**extend the differences between algorithms, and between retrieved and models XCO2, are correlated with AOD**
**and surface albedo. When such an analysis is included I recommend publication of the manuscript in AMT.**

According to your suggestion, we added a quantitative analysis about the effect of aerosols and albedo in the
discussion section from in the revised manuscript. It is also shown as follows:

[revised manuscript text omitted]

**-Other points:**

**--How accurate are the XCO2 values modeled by GEOS-CHEM? The paper would benefit from a**
**demonstration of the capability of GEOS-CHEM, for example from comparions with TCCON (albeit outside**
**the study region).**

We added comparisons of GEOS-Chem with 14 TCCON sites. The added descriptions and validation results are
shown in the revised manuscript and as follows:

We compared GEOS-Chem CO$_2$ simulations from the global model driven by CHRED with daily mean TCCON data from 14 TCCON sites (version GGG2014 data version) (Blumenstock et al., 2014; Deutscher et al., 2014; Griffith et al.,

2014a, 2014b; Hase et al., 2014; Kawakami et al., 2014; Kivi et al., 2014; Morino et al., 2014; Sherlock et al., 2014;

Sussmann et al., 2014; Warneke et al., 2014; Wennberg et al., 2014a, 2014b, 2014c). All TCCON measurements between 12

pm and 13:30 pm are used in the comparisons, where GEOS-Chem CO$_2$ profiles are taken according to the location of

TCCON stations (latitude and longitude) as well as the observing date and transformed to XCO$_2$ by convolved with the individual averaging kernel in each station as Wunch (2010) suggested. The statistics results are shown in Table 2.

**Table 2. Statistics of comparison between GEOS-Chem CO$_2$ simulations driven by CHRED and TCCON data from January 2010**
**to February 2013, which includes biases (Δ), the standard deviations (δ), the correlation coefficients (r) and valid days (days) when**
**TCCON data are available. Δ, δ and r are calculated using coincident daily mean data averaged between 12:00 pm and 13:30 pm.**

| ID | Station name | Latitude | Longitude | Δ[ppm] | δ[ppm] | r | days |
|----|--------------|----------|-----------|--------|--------|------|------|
| 1 | Sodankyla | 67.37 | 26.63 | 2.03 | 2.00 | 0.83 | 269 |
| 2 | Bialystok | 53.23 | 23.02 | 0.49 | 1.84 | 0.87 | 196 |
| 3 | Karlsruhe | 49.1 | 8.44 | 0.84 | 1.69 | 0.84 | 152 |
| 4 | Orleans | 47.97 | 2.11 | 0.44 | 1.70 | 0.85 | 223 |

| 5 | Garmisch | 47.48 | 11.06 | 0.65 | 1.64 | 0.83 | 293 |
|---|---|---|---|---|---|---|---|
| 6 | Park Falls | 45.94 | -90.27 | 1.17 | 2.14 | 0.75 | 494 |
| 7 | Lamont | 36.6 | -97.49 | -0.04 | 1.22 | 0.90 | 642 |
| 8 | Tsukuba | 36.05 | 140.12 | 1.43 | 1.66 | 0.75 | 217 |
| 9 | JPL | 34.2 | -118.18 | -1.30 | 1.15 | 0.90 | 289 |
| 10 | Saga | 33.24 | 130.29 | -0.39 | 1.65 | 0.86 | 159 |
| 11 | Izana | 28.3 | -16.48 | 0.85 | 1.04 | 0.90 | 114 |
| 12 | Darwin | -12.43 | 130.89 | 0.65 | 0.90 | 0.88 | 447 |
| 13 | Wollongong | -34.41 | 150.88 | 0.53 | 0.83 | 0.94 | 347 |
| 14 | Lauder | -45.04 | 169.68 | 0.92 | 0.42 | 0.97 | 370 |
| | Mean | | | $0.59\pm0.80$ | $1.42\pm0.50$ | | |

The results of Table 2 show that the bias ranges from -1.30 to 2.03 ppm for all TCCON sites with standard deviations of
the difference varying from 0.42 to 2.14 ppm. The mean standard deviation at the TCCON sites, a measure of the achieved
overall precision,  from using GEOS-Chem simulations driven by CHRED is $1.42\pm0.50$ ppm which is slightly different
from using GEOS-Chem simulations driven by ODIAC ($1.41\pm0.49$ ppm). Those validated results with TCCON comparing
GEOS-Chem $CO_2$ simulations driven by CHRED to that by ODIAC indicate that the GEOS-Chem $CO_2$ simulations driven
by CHRED is more likely not to change the global magnitude of $CO_2$ concentration but rather to depict fine spatial
distribution of $CO_2$ concentration in China.

**-- EMMA should be excluded from the analysis in this paper as it is not a retrieval algorithm itself but is**
**composed from the different algorithms that are also analyzed in the present study. In fact, each EMMA value is**
**the XCO2 retrieved by one of the algorithms that is closest to the median value for a given grid box. By**
**including it in this study it correlates algorithm to itself.**
We removed EMMA from the analysis according to you suggestion and the related analysis were updated in the
revised manuscript. Please refer the details to the manuscript. Please refer the details to the revised manuscript because
of difficulty in presenting it here since the changes were made across several sections.
The new analysis results for four algorithms (ACOS, NIES, OCFP, SRFP) have not changes only Table 5 (new
and old shown as below) have slight changes as EMMA is the median value among multiple algorithms including our
discussing four algorithms.
**New Table 5**
**The average of the absolute differences (ppm) and standard deviation (ppm) of the target algorithm (in column)**
**matching all other algorithms for each cell. Values in parentheses are the corresponding standard deviations.**
**The differences, which are larger than 1.5 ppm, are highlighted in bold and underlined.**

| Left longitude of cells(°E) | 80 | 85 | 90 | 95 | 100 | 105 | 110 | 115 |
|---|---|---|---|---|---|---|---|---|
| ACOS | 1.3(1.1) | 1.2(1.0) | 1.0(0.7) | 1.4(1.2) | 1.2(0.9) | 1.0(0.7) | 0.9(0.6) | 0.7(0.5) |
| NIES | 1.1(0.7) | 1.3(0.9) | 1.2(0.9) | **1.6**(1.2) | 1.1(0.8) | 1.1(0.8) | 1.1(0.8) | 0.9(0.6) |

| | 80 | 85 | 90 | 95 | 100 | 105 | 110 | 115 |
|---|---|---|---|---|---|---|---|---|
| OCFP | **1.5**(1.1) | 1.4(1.0) | 1.4(1.0) | 1.3(0.9) | 1.2(0.9) | 0.9(0.6) | 0.8(0.6) | 0.8(0.6) |
| SRFP | 1.1(0.9) | 1.2(1.0) | 1.4(1.1) | 1.2(0.9) | 1.1(0.8) | 0.9(0.6) | 1.0(0.7) | 0.8(0.5) |

**Old Table 5**

| Left longitude of cells(°E) | 80 | 85 | 90 | 95 | 100 | 105 | 110 | 115 |
|---|---|---|---|---|---|---|---|---|
| ACOS | 1.5(0.8) | 1.4(0.7) | 1.2(0.4) | 1.6(1.0) | 1.4(0.6) | 1.1(0.4) | 1.1(0.2) | 0.9(0.2) |
| NIES | 1.6(0.2) | 1.8(0.4) | 1.6(0.4) | **2.2**(0.6) | 1.6(0.3) | 1.5(0.3) | 1.5(0.3) | 1.3(0.2) |
| OCFP | **2.2**(0.6) | **2.1**(0.6) | 1.9(0.5) | 1.7(0.2) | 1.7(0.4) | 1.2(0.1) | 1.1(0.1) | 1.0(0.2) |
| SRFP | 1.3(0.5) | 1.4(0.7) | 1.6(0.8) | 1.4(0.6) | 1.3(0.5) | 1.1(0.3) | 1.2(0.4) | 1.0(0.2) |
| EMMA | 1.6(0.9) | 1.6(1.0) | 1.3(0.6) | 1.3(0.6) | 1.3(0.6) | 1.1(0.5) | 1.1(0.4) | 1.0(0.4) |

**-- A proper reference should be made to EMMA as a tool to study consistency between different algorithms, like is being done in the present study.**

Thanks for this suggestion. We will study the consistency of algorithms for EMMA in further when a proper reference is available.

**--Line 132 states: " The recommended bias corrections are applied to the collected XCO2 data from ACOS, OCFP and SRFP". What is meant here? The files for both products already contain bias corrected products. Have these been used?**

This is our incorrect expression. Modified to: "The collected XCO2 data from ACOS, OCFP and SRFP are products after bias correction." .

**-- Line 364 stated:" while Aerosol Optical Depth (AOD) is greatly affected by high surface albedo because of the optical lengthening effect.". What is meant here? AOD is not affected by surface albedo.**

It is our incorrect expression. Modified to: "while  estimations of Aerosol Optical Depth (AOD) in GOSAT full physics CO2 retrieval algorithms are greatly affected by high surface albedo because of atmospheric multiple scattering of light and the optical lengthening effect" .

**-- The additional analysis of the new ACOS V7.3 product is confusing. It should either be used in the full analysis or the discussion should be shortened by only stating to what extend the conclusions would be different if the ACOS V7.3 product would have been used. The more detailed analysis could be moved to an appendix.**

We shortened the part on the new version of ACOS, and moved part of it to an appendix according to your suggestion. Please refer the details to the revised manuscript. We use ACOS V3.5 instead of ACOS V7.3, the more recently released products, in the analysis because we considered that (1) ACOS V3.5 have been being currently used in our studying group; (2) as described in reference[GES DISC, 2017], which says, T*he retrieval algorithm used to create the Build 7 ACOS data product is consistent with that used to create the OCO-2 v7.3 data product. This will allow comparison of the ACOS and OCO-2 data without having to consider algorithm differences,* ACOS V7.3 are not exactly the newer version of ACOS products.

---

## Author Comment (AC3) · 9 Jan 2018

[revised manuscript text omitted]
 | | -1.4(1.2) | **-2.6**(1.2) | -0.5(1.2) |  | | **-1.6**(1.6) | **-2.0**(1.1) | -0.2(1.2) |  |
| NIES | 80 °E | | -0.9(1.4) | 1.1(1.4) |  | 100 °E | | -0.4(1.4) | 1.4(1.5) |  |
| OCFP | | | | **2.0**(1.2) |  | | | | **1.7**(1.3) |  |
| SRFP | | | | |  | | | | |  |
| ACOS | | **-2.0**(1.3) | **-1.9**(1.2) | -0.1(1.2) |  | | **-1.6**(1.3) | -0.6(1.4) | 0.2(1.2) |  |
| NIES | 85 °E | | -0.4(1.6) | 1.5(1.3) |  | 105 °E | | 0.2(1.5) | 1.2(1.3) |  |
| OCFP | | | | **2.3**(1.4) |  | | | | 1.0(1.3) |  |
| SRFP | | | | |  | | | | |  |
| ACOS | | -1.2(1.1) | **-1.7**(1.1) | 0.8(1.4) |  | | -1.2(1.3) | -0.9(1.4) | 0.0(1.4) |  |
| NIES | 90 °E | | -0.8(1.4) | **2.0**(1.4) |  | 110 °E | | 0.7(1.3) | 1.5(1.6) |  |
| OCFP | | | | **2.4**(1.5) |  | | | | 0.5(1.2) |  |
| SRFP | | | | |  | | | | |  |
| ACOS | | **-3.0**(1.1) | -0.9(1.7) | -0.3(1.2) |  | | -0.6(1.3) | 0.1(1.0) | -0.1(1.0) |  |
| NIES | 95 °E | | 0.5(2.1) | 1.3(2.0) |  | 115 °E | | 0.8(1.5) | 0.9(1.3) |  |
| OCFP | | | | **1.8**(1.6) |  | | | | 0.2(1.3) |  |

[revised manuscript text omitted]

---

## Author Response (AR1)

**A list of all relevant changes**

Dear editor:

Thanks for your work and the referees' contributions to the improvement of our paper. We are very grateful for that. I write to give you a general picture of the major revisions we have made as the referees suggested.

(1)     We added a quantitative analysis of the effect of aerosols and albedo on differences between retrieved and models XCO2 as well as differences in retrieved XCO2 between different algorithms as the two referees suggested.

(2)     We removed EMMA, one of the algorithms, from the analysis in this paper as the two referees suggested. And we also revised the related analysis results.

(3)     We shortened the part about ACOS V7.3 and move part of it to the appendix.

(4)     According to the referees' suggestions, we revised our conclusion and analysis results in Table 7 so as to be more concise and conclusive.

(5)     We corrected improper English in the paper.

Best regards,

Nian

**Responses to referee #1**

Responses to Anonymous Referee #1 on the manuscript of "Regional uncertainty of GOSAT XCO2 retrievals in China: Quantification and attribution"

Thank you for your suggestions and valuable comments very much. We have fully considered all your comments, and carried out our revision and improved our manuscript accordingly. The item-by-item response to the specific comments is as follows (referee's comments in **red** and our response in **black**).

**Referee #1: general:**

**-The paper is interesting to the CO2 remote sensing community although in the end it stays rather inconclusive. The reason is that there is no absolute reference for the true XCO2 in this study. The conclusions that are being drawn are based on (in-) consistency between different retrieval algorithms and comparison to the GEOS-CHEM model and are hence to large extend speculative.**

For inconclusive problem as you point out, we revised our analysis results concluded in Table 7. In this study, we aim to reveal regional uncertainty of GOSAT XCO2 retrievals via comparison and evaluation of consistence of multi-algorithms for GOSAT observations, and probe the reason why performances of $XCO_2$ from multi- algorithms are different in same regions. Our results are expected to give a reliable and valuable reference for application of XCO2 data in detection of carbon source and sink at a regional scale, e.g. the result gotten by our analysis, the better consistence of XCO2 from four algorithms (ACOS, NIES, OCFP, SRFP) in Eastern China with large anthropogenic CO2 emissions, can promote us to detect the anthropogenic enhancement of CO2 concentrations using these XCO2 data with confidence, and the result, the existing problems in deserts likely influenced by albedo and AOD, is expected to get attentions and improvement.

[revised manuscript text omitted]

**-Other points:**

**--How accurate are the XCO2 values modeled by GEOS-CHEM? The paper would benefit from a demonstration of**

**the capability of GEOS-CHEM, for example from comparions with TCCON (albeit outside the study region).**

We added comparisons of GEOS-Chem with 14 TCCON sites. The added descriptions and validation results are shown in the revised manuscript and as follows:

We compared GEOS-Chem $CO_2$ simulations from the global model driven by CHRED with daily mean TCCON data from 14 TCCON sites (version GGG2014 data version) (Blumenstock et al., 2014; Deutscher et al., 2014; Griffith et al.,

2014a, 2014b; Hase et al., 2014; Kawakami et al., 2014; Kivi et al., 2014; Morino et al., 2014; Sherlock et al., 2014;

Sussmann et al., 2014; Warneke et al., 2014; Wennberg et al., 2014a, 2014b, 2014c). All TCCON measurements between 12

pm and 13:30 pm are used in the comparisons, where GEOS-Chem $CO_2$ profiles are taken according to the location of

TCCON stations (latitude and longitude) as well as the observing date and transformed to $XCO_2$ by convolved with the individual averaging kernel in each station as Wunch (2010) suggested. The statistics results are shown in Table 5.

**Table 2. Statistics of comparison between GEOS-Chem $CO_2$ simulations driven by CHRED and TCCON data from January 2010**

**to February 2013, which includes biases (Δ), the standard deviations (δ), the correlation coefficients (r) and valid days (days) when**

**TCCON data are available. Δ, δ and r are calculated using coincident daily mean data averaged between 12:00 pm and 13:30 pm.**

| ID | Station name | Latitude | Longitude | Δ[ppm] | δ[ppm] | r | days |
|----|----|----|----|----|----|----|----|

| | | | | | | | |
|---|---|---|---|---|---|---|---|
| 1 | Sodankyla | 67.37 | 26.63 | 2.03 | 2.00 | 0.83 | 269 |
| 2 | Bialystok | 53.23 | 23.02 | 0.49 | 1.84 | 0.87 | 196 |
| 3 | Karlsruhe | 49.1 | 8.44 | 0.84 | 1.69 | 0.84 | 152 |
| 4 | Orleans | 47.97 | 2.11 | 0.44 | 1.70 | 0.85 | 223 |
| 5 | Garmisch | 47.48 | 11.06 | 0.65 | 1.64 | 0.83 | 293 |
| 6 | Park Falls | 45.94 | -90.27 | 1.17 | 2.14 | 0.75 | 494 |
| 7 | Lamont | 36.6 | -97.49 | -0.04 | 1.22 | 0.90 | 642 |
| 8 | Tsukuba | 36.05 | 140.12 | 1.43 | 1.66 | 0.75 | 217 |
| 9 | JPL | 34.2 | -118.18 | -1.30 | 1.15 | 0.90 | 289 |
| 10 | Saga | 33.24 | 130.29 | -0.39 | 1.65 | 0.86 | 159 |
| 11 | Izana | 28.3 | -16.48 | 0.85 | 1.04 | 0.90 | 114 |
| 12 | Darwin | -12.43 | 130.89 | 0.65 | 0.90 | 0.88 | 447 |
| 13 | Wollongong | -34.41 | 150.88 | 0.53 | 0.83 | 0.94 | 347 |
| 14 | Lauder | -45.04 | 169.68 | 0.92 | 0.42 | 0.97 | 370 |
| | Mean | | | $0.59\pm0.80$ | $1.42\pm0.50$ | | |

The results of Table 5 show that the bias ranges from -1.30 to 2.03 ppm for all TCCON sites with standard deviations of the difference varying from 0.42 to 2.14 ppm. The mean standard deviation at the TCCON sites, a measure of the achieved overall precision, from using GEOS-Chem simulations driven by CHRED is $1.42\pm0.50$ ppm which is slightly different from using GEOS-Chem simulations driven by ODIAC ($1.41\pm0.49$ ppm). Those validated results with TCCON comparing GEOS-Chem $CO_2$ simulations driven by CHRED to that by ODIAC indicate that the GEOS-Chem $CO_2$ simulations driven by CHRED is more likely not to change the global magnitude of $CO_2$ concentration but rather to depict fine spatial distribution of $CO_2$ concentration in China.

**-- EMMA should be excluded from the analysis in this paper as it is not a retrieval algorithm itself but is composed from the different algorithms that are also analyzed in the present study. In fact, each EMMA value is the XCO2 retrieved by one of the algorithms that is closest to the median value for a given grid box. By including it in this study it correlates algorithm to itself.**

We removed EMMA from the analysis according to you suggestion and the related analysis were updated in the revised manuscript. Please refer the details to the manuscript. Please refer the details to the revised manuscript because of difficulty in presenting it here since the changes were made across several sections.

The new analysis results for four algorithms (ACOS, NIES, OCFP, SRFP) have not changes only Table 5 (new and old shown as below) have slight changes as EMMA is the median value among multiple algorithms including our discussing four algorithms.

**New Table 5**

**The average of the absolute differences (ppm) and standard deviation (ppm) of the target algorithm (in column)**

**matching all other algorithms for each cell. Values in parentheses are the corresponding standard deviations. The**

**differences, which are larger than 1.5 ppm, are highlighted in bold and underlined.**

| Left longitude of cells( °E) | 80 | 85 | 90 | 95 | 100 | 105 | 110 | 115 |
|---|---|---|---|---|---|---|---|---|
| ACOS | 1.3(1.1) | 1.2(1.0) | 1.0(0.7) | 1.4(1.2) | 1.2(0.9) | 1.0(0.7) | 0.9(0.6) | 0.7(0.5) |
| NIES | 1.1(0.7) | 1.3(0.9) | 1.2(0.9) | **1.6**(1.2) | 1.1(0.8) | 1.1(0.8) | 1.1(0.8) | 0.9(0.6) |
| OCFP | **1.5**(1.1) | 1.4(1.0) | 1.4(1.0) | 1.3(0.9) | 1.2(0.9) | 0.9(0.6) | 0.8(0.6) | 0.8(0.6) |
| SRFP | 1.1(0.9) | 1.2(1.0) | 1.4(1.1) | 1.2(0.9) | 1.1(0.8) | 0.9(0.6) | 1.0(0.7) | 0.8(0.5) |

**Old Table 5**

| Left longitude of cells( °E) | 80 | 85 | 90 | 95 | 100 | 105 | 110 | 115 |
|---|---|---|---|---|---|---|---|---|
| ACOS | 1.5(0.8) | 1.4(0.7) | 1.2(0.4) | 1.6(1.0) | 1.4(0.6) | 1.1(0.4) | 1.1(0.2) | 0.9(0.2) |
| NIES | 1.6(0.2) | 1.8(0.4) | 1.6(0.4) | **2.2**(0.6) | 1.6(0.3) | 1.5(0.3) | 1.5(0.3) | 1.3(0.2) |
| OCFP | **2.2**(0.6) | **2.1**(0.6) | 1.9(0.5) | 1.7(0.2) | 1.7(0.4) | 1.2(0.1) | 1.1(0.1) | 1.0(0.2) |
| SRFP | 1.3(0.5) | 1.4(0.7) | 1.6(0.8) | 1.4(0.6) | 1.3(0.5) | 1.1(0.3) | 1.2(0.4) | 1.0(0.2) |
| EMMA | 1.6(0.9) | 1.6(1.0) | 1.3(0.6) | 1.3(0.6) | 1.3(0.6) | 1.1(0.5) | 1.1(0.4) | 1.0(0.4) |

**-- A proper reference should be made to EMMA as a tool to study consistency between different algorithms, like is**

**being done in the present study.**

Thanks for this suggestion. We will study the consistency of algorithms for EMMA in further when a proper reference is available.

**--Line 132 states: " The recommended bias corrections are applied to the collected XCO2 data from ACOS, OCFP**

**and SRFP". What is meant here? The files for both products already contain bias corrected products. Have these**

**been used?**

This is our incorrect expression. Modified to: "The collected XCO2 data from ACOS, OCFP and SRFP are products after bias correction." .

**-- Line 364 stated:" while Aerosol Optical Depth (AOD) is greatly affected by high surface albedo because of the**

**optical lengthening effect.". What is meant here? AOD is not affected by surface albedo.**

It is our incorrect expression. Modified to: "while  estimations of Aerosol Optical Depth (AOD) in GOSAT full physics

CO2 retrieval algorithms are greatly affected by high surface albedo because of atmospheric multiple scattering of light and the optical lengthening effect" .

**-- The additional analysis of the new ACOS V7.3 product is confusing. It should either be used in the full analysis or**

**the discussion should be shortened by only stating to what extend the conclusions would be different if the ACOS**

**V7.3 product would have been used. The more detailed analysis could be moved to an appendix.**

We shortened the part on the new version of ACOS, and moved part of it to an appendix according to your suggestion. Please refer the details to the revised manuscript. We use ACOS V3.5 instead of ACOS V7.3, the more recently released products, in the analysis because we considered that (1) ACOS V3.5 have been being currently used in our studying group; (2) as described in reference[GES DISC, 2017], which says, T*he retrieval algorithm used to create the Build 7 ACOS data product is consistent with that used to create the OCO-2 v7.3 data product. This will allow comparison of the ACOS and OCO-2 data without having to consider algorithm differences,* ACOS V7.3 are not exactly the newer version of ACOS products.

**Responses to referee #2**

Responses to Anonymous Referee #2 on the manuscript of "Regional uncertainty of GOSAT XCO2 retrievals in China: Quantification and attribution"

Thank you for your suggestions and valuable comments very much. We have fully considered all your comments, and carried out our revision and improved our manuscript accordingly. The item-by-item response to the specific comments is as follows (referee's comments in **red** and our response in **black**).

**Referee #2:**

**Major points : See the comments from the other reviewer :**

**- EMMA should be left out as it is the combined product of all other retrieval products shown**

We removed EMMA from the analysis according to you suggestion and the related analysis were updated in the revised manuscript. Please refer the details to the revised manuscript because of difficulty in presenting it here since the changes were made across several sections.

The new analysis results for four algorithms (ACOS, NIES, OCFP, SRFP) have not changes only Table 5 (new and old shown as below) have slight changes as EMMA is the median value among multiple algorithms including our discussing four algorithms.

**New Table 5**

**The average of the absolute differences (ppm) and standard deviation (ppm) of the target algorithm (in column) matching all other algorithms for each cell. Values in parentheses are the corresponding standard deviations. The differences, which are larger than 1.5 ppm, are highlighted in bold and underlined.**

| Left longitude of cells( °E) | 80 | 85 | 90 | 95 | 100 | 105 | 110 | 115 |
|---|---|---|---|---|---|---|---|---|
| ACOS | 1.3(1.1) | 1.2(1.0) | 1.0(0.7) | 1.4(1.2) | 1.2(0.9) | 1.0(0.7) | 0.9(0.6) | 0.7(0.5) |
| NIES | 1.1(0.7) | 1.3(0.9) | 1.2(0.9) | **1.6**(1.2) | 1.1(0.8) | 1.1(0.8) | 1.1(0.8) | 0.9(0.6) |
| OCFP | **1.5**(1.1) | 1.4(1.0) | 1.4(1.0) | 1.3(0.9) | 1.2(0.9) | 0.9(0.6) | 0.8(0.6) | 0.8(0.6) |
| SRFP | 1.1(0.9) | 1.2(1.0) | 1.4(1.1) | 1.2(0.9) | 1.1(0.8) | 0.9(0.6) | 1.0(0.7) | 0.8(0.5) |

**Old Table 5**

| Left longitude of cells( °E) | 80 | 85 | 90 | 95 | 100 | 105 | 110 | 115 |
|---|---|---|---|---|---|---|---|---|
| ACOS | 1.5(0.8) | 1.4(0.7) | 1.2(0.4) | 1.6(1.0) | 1.4(0.6) | 1.1(0.4) | 1.1(0.2) | 0.9(0.2) |
| NIES | 1.6(0.2) | 1.8(0.4) | 1.6(0.4) | **2.2**(0.6) | 1.6(0.3) | 1.5(0.3) | 1.5(0.3) | 1.3(0.2) |
| OCFP | **2.2**(0.6) | **2.1**(0.6) | 1.9(0.5) | 1.7(0.2) | 1.7(0.4) | 1.2(0.1) | 1.1(0.1) | 1.0(0.2) |
| SRFP | 1.3(0.5) | 1.4(0.7) | 1.6(0.8) | 1.4(0.6) | 1.3(0.5) | 1.1(0.3) | 1.2(0.4) | 1.0(0.2) |
| EMMA | 1.6(0.9) | 1.6(1.0) | 1.3(0.6) | 1.3(0.6) | 1.3(0.6) | 1.1(0.5) | 1.1(0.4) | 1.0(0.4) |

**- Shorten the part on the new version of ACOS, or use only the new version data**

We shortened the part on the new version of ACOS, and moved part of it to the appendix according to your suggestion. Please refer the details to the revised manuscript. We use ACOS V3.5 instead of ACOS V7.3, the more recently released products, in the analysis because we considered that (1) ACOS V3.5 have been being currently used in our studying group; (2) as described in reference[GES DISC, 2017], which says, T*he retrieval algorithm used to create the Build 7 ACOS data product is consistent with that used to create the OCO-2 v7.3 data product. This will allow comparison of the ACOS and*

*OCO-2 data without having to consider algorithm differences,* ACOS V7.3 is not exactly the newer version of ACOS
products.

**- Provide a more quantitative analysis of the effect of aerosols and albedo on the observed differences between**
**different algorithms**
According to your suggestion, we added a quantitative analysis about the effect of aerosols and albedo in the discussion
section in the revised manuscript and presented it here:
We discussed the influences of albedo and AOD on $XCO_2$ retrievals from ACOS, NIES, OCFP and SRFP in further.
Fig. 14 plots the scatters of albedo and AOD with the differences between GEOS-XCO2 data (created in section 3.1) to
$XCO_2$ retrievals, hereafter referred to as $dmXCO_2$, for ACOS, NIES, OCFP and SRFP. The albedo data obtained from
GLASS02B06 is used for OCFP as there are no albedo data available from its released data product.
Fig. 14 shows that $dmXCO_2$ of both ACOS and NIES demonstrate a slightly decreasing trend with albedo whereas
slightly increasing trend with AOD. The $dmXCO_2$ of ACOS tend to be larger in 80 °E -90 °E of deserts with high albedo than
that in other regions. The dmXCO2 of OCFP demonstrate a clear decreasing trend with albedo and AOD comparing to the
other algorithms. The $dmXCO_2$ of SRFP basically does not show a clearly dependence on either albedo or AOD. We further
investigated the standard deviation of $dmXCO_2$ by a variation of the bin-to-bin $dmXCO_2$ with albedo and AOD. $dmXCO_2$ is
averaged by surface albedo within 0.05 albedo bins and AOD within 0.05 AOD bins, respectively. The standard deviation of
the mean $dmXCO_2$ in each 0.05 albedo (AOD) bins, i.e. a measure of the bin-to-bin $dmXCO_2$, is calculated. It is found that
the dmXCO2 for the four algorithms change with both albedo and AOD in bin-to-bin. In the whole study area, the standard
deviation in albedo is the largest for OCFP, up to 0.7 ppm, while that is smaller from ACOS, NIES and SRFP, 0.4 ppm、0.3
ppm and  0.2 ppm, respectively. The standard deviation of $dmXCO_2$ in AOD is larger for SRFP (0.5 ppm) than those for
ACOS (0.2 ppm), NIES (0.3 ppm) and OCFP (0.4 ppm). Viewing to the deserts (80 °E -90 °E), the standard deviation  in
albedo is the largest from NIES ( 1.5 ppm),  and the smallest from OCFP (0.2 ppm) while they are 1.0 ppm and 0.5 ppm for
ACOS and SRFP, respectively. The standard deviations in AOD, however, are similar (0.2-0.4 ppm) in this area. As a result,
OCFP tend to be more sensitive to albedo and AOD compared to other algorithms. In the deserts, NIES are the most
sensitive $XCO_2$ retrievals to surface albedo and OCFP the least.

[Figure]

**Fig. 3: Scatter plots of the differences (dmXCO$_2$) between GEOS-XCO$_2$ to ACOS, NIES, OCFP and SRFP respectively, with respect to albedo (the upper panels) and AOD (the lower panels). Colored points represent the data from different cells: red-[80°E, 105°E], black-[105°E, 120°E] in the study latitude zone [37°N, 42°N]. Colored solid lines display the corresponding linear regression trend line for the total points. Albedo and AOD are extracted from data products of the retrieval algorithms except albedo data in OCFP in which GLASS data are used.**

Figure Fig. 15, moreover, demonstrates the influence of albedo and AOD on the standard deviation (STD) of XCO$_2$ from four algorithms at the same footprints (timely in the same day, geometrically located within ±0.01° in space). Averaged albedo (the left panels) and AOD (the right panels) of the four algorithms are used whereas the averaged albedo is obtained only using three attached albedo in the algorithms except OCFP.

The increasing trends of STD with both albedo and AOD can be seen from Fig. 15. The mean STD is 1.3 ppm in the western cells (80°E -90°E) where albedo is mostly within 0.25-0.35. This STD is lightly larger than that (1.0ppm) in eastern cells (90°E-120E°) where albedo is comparatively smaller (mostly within 0.15-0.25). It is found from the statistics presented in Fig. 15 that the correlation coefficients of STD with albedo and that with AOD is almost the same (both are 0.3) for all the data. Particular influence from albedo in desert over the western cells can be clearly observed. These results indicate that the inconsistency of XCO$_2$ retrievals from four algorithms tend to be increase with the enlargements of albedo and AOD so as to imply that uncertainty of satellite-retrieved XCO$_2$ should be mostly alerted with the elevations of albedo and AOD.

[Figure]

**Fig. 4: Scatter plots of the standard deviation (STD) of XCO₂ from the four algorithms to albedo (the left panel) and AOD (the**
**right panel). Colored points represent different cells: red-[80 °E, 105 °E], black-[105 °E, 120 °E] in the latitude zone [37 °N, 42 °N].**
**Colored solid lines display the corresponding linear regression trend line for the scatter plots with the regression slope (a) and the**
**correlation coefficient (r) also presented. n is the number of samples. Albedo is the mean surface albedo in 0.75-um band from the**
**three algorithms including ACOS, NIES and SRFP. AOD is the mean AOD in 0.75-um band from the four algorithms.**

**- Provide some clear evidence of performance of GEOS-Chem wrt total column XCO2**

We added comparisons of GEOS-Chem with 14 TCCON sites. The added descriptions and validation results are shown in the revised manuscript and as follows:

We compared GEOS-Chem $CO_2$ simulations from the global model driven by CHRED with daily mean TCCON data from 14 TCCON sites (version GGG2014 data version) (Blumenstock et al., 2014; Deutscher et al., 2014; Griffith et al.,

2014a, 2014b; Hase et al., 2014; Kawakami et al., 2014; Kivi et al., 2014; Morino et al., 2014; Sherlock et al., 2014;

Sussmann et al., 2014; Warneke et al., 2014; Wennberg et al., 2014a, 2014b, 2014c). All TCCON measurements between 12

pm and 13:30 pm are used in the comparisons, where GEOS-Chem $CO_2$ profiles are taken according to the location of

TCCON stations (latitude and longitude) as well as the observing date and transformed to $XCO_2$ by convolved with the individual averaging kernel in each station as Wunch (2010) suggested. The statistics results are shown in Table 5.

**Table 3. Statistics of comparison between GEOS-Chem $CO_2$ simulations driven by CHRED and TCCON data from January 2010**
**to February 2013, which includes biases (Δ), the standard deviations (δ), the correlation coefficients (r) and valid days (days) when**
**TCCON data are available. Δ, δ and r are calculated using coincident daily mean data averaged between 12:00 pm and 13:30 pm.**

| ID | Station name | Latitude | Longitude | Δ[ppm] | δ[ppm] | r | days |
|----|--------------|----------|-----------|--------|--------|------|------|
| 1 | Sodankyla | 67.37 | 26.63 | 2.03 | 2.00 | 0.83 | 269 |
| 2 | Bialystok | 53.23 | 23.02 | 0.49 | 1.84 | 0.87 | 196 |
| 3 | Karlsruhe | 49.1 | 8.44 | 0.84 | 1.69 | 0.84 | 152 |
| 4 | Orleans | 47.97 | 2.11 | 0.44 | 1.70 | 0.85 | 223 |
| 5 | Garmisch | 47.48 | 11.06 | 0.65 | 1.64 | 0.83 | 293 |
| 6 | Park Falls | 45.94 | -90.27 | 1.17 | 2.14 | 0.75 | 494 |
| 7 | Lamont | 36.6 | -97.49 | -0.04 | 1.22 | 0.90 | 642 |
| 8 | Tsukuba | 36.05 | 140.12 | 1.43 | 1.66 | 0.75 | 217 |

| 9 | JPL | 34.2 | -118.18 | -1.30 | 1.15 | 0.90 | 289 |
| 10 | Saga | 33.24 | 130.29 | -0.39 | 1.65 | 0.86 | 159 |
| 11 | Izana | 28.3 | -16.48 | 0.85 | 1.04 | 0.90 | 114 |
| 12 | Darwin | -12.43 | 130.89 | 0.65 | 0.90 | 0.88 | 447 |
| 13 | Wollongong | -34.41 | 150.88 | 0.53 | 0.83 | 0.94 | 347 |
| 14 | Lauder | -45.04 | 169.68 | 0.92 | 0.42 | 0.97 | 370 |
| | Mean | | | $0.59\pm0.80$ | $1.42\pm0.50$ | | |

The results of Table 5 show that the bias ranges from -1.30 to 2.03 ppm for all TCCON sites with standard deviations of the difference varying from 0.42 to 2.14 ppm. The mean standard deviation at the TCCON sites, a measure of the achieved overall precision, from using GEOS-Chem simulations driven by CHRED is $1.42\pm0.50$ ppm which is slightly different from using GEOS-Chem simulations driven by ODIAC ($1.41\pm0.49$ ppm). Those validated results with TCCON comparing

GEOS-Chem $CO_2$ simulations driven by CHRED to that by ODIAC indicate that the GEOS-Chem $CO_2$ simulations driven by CHRED is more likely not to change the global magnitude of $CO_2$ concentration but rather to depict fine spatial distribution of $CO_2$ concentration in China.

**Minor : Textual suggestions :**

**-p.2 line 46 : I think you should leave out TanSat in that particular sentence as that instrument has not yet**

**contributed to a better understanding of…as far as I know.**

Yes, TanSat have not produces XCO2 data available as to its some problems as you know. We removed the description of TanSat in the revised manuscript.

**-p.3 line 85-86 : rephrase 'that trend …to east' because unclear what is meant**

Modified to: " there are anthropogenic emissions increasing from west to east." in line 83.

**-p.9 GLASS albedo is used. For which wavelength is this albedo?**

It is broadband albedo product rather than albedo in narrow bands. The following was added: " GLASS02B06 is a daily land-surface shortwave (300-3000nm) broadband albedo product in temporal resolution of eight days.".

**-table 2. Add to the table caption : All biases > 1 ppm are underlined.**

We added it in the caption of table 3, which is the previous table 2. The caption is modified to: "The biases (ppm) and their standard deviations (ppm) of the four algorithms vs GEOS-Chem in each cell, where the upper line indicates bias (the corresponding standard deviations in parenthesis) for each algorithm vs GEOS-Chem and the lower line is the available number of used samples. The biases, larger than 1 ppm, are highlighted in bold and underlined." in the revised manuscript.

**-Change 'the values in parentheses are the biases and their …" → "the values are the biases and –in parentheses-**

**their…'**

We revised this incorrect description, which also refers to the caption of table 3, in the revised manuscript. If you have read the last item, the following five lines can be skipped.

The caption is modified to:"The biases (ppm) and standard deviation (ppm) of the four algorithms vs GEOS-Chem in
each cell, where the upper line indicates bias(the standard deviations) for each algorithm vs GEOS-Chem and the lower line
is the number of used samples. The biases, larger than 1 ppm, are highlighted in bold and underlined." in the revised
manuscript.
**-Table 3 table caption. What are the underlined values ?**
They are differences (ppm) larger than 1.5 ppm between two algorithms (column algorithm minus row algorithm) for
each cell.
The caption of Table 4, which is the previous table 3, was modified to: "Differences (ppm) between two algorithms
(column algorithm minus row algorithm) and the standard deviation (ppm) for each cell, where values in parentheses are the
corresponding standard deviations. The differences, larger than 1.5 ppm, are highlighted in bold and underlined."  in the
revised manuscript.
**p.18 line 350 ('To summarize the quantification…SRFP') : I do not understand this sentence given the data.**
Thank you for pointing it out. This sentence has been deleted in the revised manuscript because we are also aware that
this sentence makes the results confusing.
**-Fig. 8 Figure caption 'and the differences of detrended…. and GEOS-Chem' should that be '… with GOES-Chem' ?**
Corrected. Modified to :" The spatial (in the study latitude band) and temporal (in seasons) changing patterns of
detrended XCO2 from ACOS, NIES, OCFP, SRFP retrievals and GEOS-Chem simulations (left) and the differences of
detrended XCO2 to GEOS-Chem for ACOS, NIES, OCFP and SRFP."
**-p.21 line 423/424 I do not understand the sentence 'No bias was found …R2=0.77' based on what I see in Table 6.**
**Also it is not consistent with what is written in line429/430.**
It is our incorrect expression. The results that no bias was found in ACOS V7.3 from GEOS-Chem with a standard
deviation of 1.6 ppm and R2=0.77, is for the whole study area. The original Line 429/430 which states, *"It can also be found*
*from Table 6 that the bias of ACOS V7.3 relative to GEOS-Chem is within 0.3 ppm but above 1.3 ppm, in cells east and west*
*of 90°E, respectively."*, is focused on the regional performance.
The sentence has been modified to:" No bias was found in ACOS V7.3 from GEOS-Chem with a standard deviation of
1.6 ppm and R2 of 0.77 in the whole study area." in the appendix.
**-p. 23, line 462 results above → results described above**
Corrected.

**Marked-up manuscript version**

[revised manuscript text omitted]